# Critical evaluation of *in situ* analyses for the characterisation of red pigments in rock paintings: A case study from El Castillo, Spain

Laure Dayet[1]*, Francesco d'Errico[2,3], Marcos García Diez[4], João Zilhão[5,6,7]

1 CNRS-Université Toulouse Jean Jaurès, UMR5608 Travaux et Recherches Archéologiques sur les Cultures, les Espaces et les Sociétés, Maison de la Recherche, France, 2 CNRS-UMR 5199 PACEA, De la Préhistoire à l'Actuel: Culture, Environnement et Anthropologie, University of Bordeaux, France, 3 Centre for Early Sapiens Behaviour, University of Bergen, Bergen, Norway, 4 Departament of Prehistory, Ancient History and Archaeology, Complutense University of Madrid (UCM), Madrid, Spain, 5 Departament de Prehistòria, Història Antiga i Arqueologia (SERP; Grup de Recerca SGR2014-00108), University of Barcelona, Barcelona, Spain, 6 Institució Catalana de Recerca i Estudis Avançats (ICREA), Barcelona, Spain, 7 UNIARQ–Centro de Arqueologia da Universidade de Lisboa, Faculdade de Letras de Lisboa, Universidade de Lisboa, Lisboa, Portugal

* laure.dayet@gmail.com

**Data Availability Statement:** All relevant data are within the paper and its Supporting Information files.

## Abstract

Paint technology, namely paint preparation and application procedures, is an important aspect of painting traditions. With the expansion of archaeometric studies and *in situ* non-destructive analytical methods, a renewal of technological studies is being observed in rock art. *In situ* analyses have several limitations that are widely discussed in the literature, however. It is not yet clear whether they provide accurate information on paint technology, except under certain conditions. Here, we evaluated digital microscopic and pXRF *in situ* analyses for the characterisation of a large set of red and yellow paintings from the El Castillo cave, Cantabria, Spain. We have set experiments and used statistical methods to identify differences between paint components and determine factors impacting pXRF measurements. We found that the compositional heterogeneity of the paintings' environment, especially variations in secondary deposits, was responsible for most of the differences observed between the pXRF signals recorded on the paintings. We concluded that the El Castillo cave environment is not suitable for non-destructive technological studies, but that more favourable contexts might exist. Following previous works and our own results, we advocate a combination of both *in situ* and laboratory invasive analyses for the study of paint composition and paint technology. Our research protocol, based on the comparison of rock paintings, their substrate, experimental paintings and Fe-normalisation of the signals can improve the reliability of pXRF results. We also propose to include more systematic characterisation of rock wall heterogeneity and the use of microscopic analyses in non-destructive approaches.

**Funding:** The work of Laure Dayet was supported by a grant from the European Research Council (FP7/2007/2013, TRACSYMBOLS 249587). The work of Francesco d'Errico is also supported by the Programme Talents and the Grand Programme de Recherche Human Past of the University of Bordeaux Initiative of Excellence, and the Research Council of Norway through its Centres of Excellence funding scheme, SFF Centre for Early Sapiens Behaviour (SapienCE), project number 262618.

**Competing interests:** The authors have declared that no competing interests exist.

# 1. Introduction

Palaeolithic rock art is a key feature of human evolution, providing information on humans' language, abstract thoughts, symbolic behaviour, ontological world and social organisation. Traditionally, rock art studies have focused on chronology, theme and style. With recent developments in archaeometric techniques, however, interest in paint technology, namely paint preparation and application procedures, has increased considerably (see e.g. [1–15]). *In-situ* analyses performed with portable X-ray fluorescence (pXRF) and Raman spectroscopy equipment are becoming widespread for the study of paint materials and painting techniques. They allow the paintings to be conserved, unlike more conventional laboratory analyses requiring sampling. However, conservation is not the only issue that must be taken into account. The reliability of the results also needs in depth evaluation. It is therefore necessary to assess the limitations of *in situ* analyses more effectively, in order to determine the extent to which they provide valid knowledge on paint technology, and need to be backed up by sampling procedures.

Over the last decade, *in situ* spectroscopic Raman and pXRF analyses have provided interesting results on the pigment families composing various open-air rock paintings and paintings associated with portable art around the world (see e.g. [2, 12, 13, 16–29]). Successful results have also been obtained on some European cave paintings, despite great difficulties gaining access to the painted panels [1, 30–36]. Microscopic *in-situ* analyses have been used less frequently in recent studies, but they do give significant information on paint components and painting application techniques [12, 37, 38]. However, the identification of complex paint recipes or 'paint pots' with these methods remains challenging, especially for red iron oxides, the most common family of rock art pigments. How *in-situ* analyses can be used to identify more than a pigment family and how they can provide consistent information on paint technology remains unclear [8, 12, 20, 39]. Recent work shows that pXRF elemental composition of the paintings is not directly comparable with the pXRF elemental composition of colouring material used to make them [13]. Yet most of the time rock art pigment composition studies operate under permits that restrict analyses to non-destructive techniques and pXRF represents the only method of data acquisition allowed.

The red paintings from El Castillo (Cantabria, Spain), inscribed on the UNESCO World Heritage List in 2008, represent an ideal case study to address this issue. Various figures are painted in red on the walls of this important cave art site, including deer, bison, tectiforms, sets of negative hand stencils, juxtaposed disks, and various other abstract representations [40–42]. Uranium-series disequilibrium dates of calcite deposits on the paintings have shown that these representations extend back to at least the Aurignacian period and possibly the late Mousterian, with minimum ages of 41.4±0.6 ka for a red disk, 37.6±0.3 ka for a hand stencil, and 35.7±0.3 ka for a red disk (95.4% probability intervals; [43]). Other Uranium-series disequilibrium and radiocarbon dates have shown that the paintings were made over a long time period from the Aurignacian (or earlier) to the Magdalenian (12–17 ka; [43–45]). Recently, interdisciplinary analyses were carried out on 11 panels of red disks and micro-samples were taken from four of them [4]. The results showed that at least two paint 'recipes' were used, with different grinding intensities and mineralogical compositions. These two different 'recipes' were applied with different blowing techniques in different parts of the cave, suggesting the panels of disks were not all made at the same time. The technological and chronological relationships between the red disks and other figures remain unknown.

In this paper, we present *in situ* analyses of a wide range of paintings from El Castillo, including the red disks. The aims of this study were to 1) test pXRF and microscopic *in-situ* analytical methods on El Castillo red paintings, 2) evaluate their reliability for the

characterisation of pictorial layers in a cave context, 3) search for significant compositional variations between different red figures from El Castillo. To achieve this, we carried out microscopic examination and pXRF analyses of several red paintings and one yellow painting for comparison purposes, and used experimental paintings and statistical analyses to assess the reliability of the results obtained on the El Castillo paintings. The interest and limitations pXRF analysis has already been discussed for other archaeological applications (see e.g. [46–50]) but no in-depth studies have been carried out in the critical domain of painted rock art. The approach we developed, which until now remained untried, was of critical importance in the revelation of the origin of the compositional differences observed on the El Castillo paintings that we were able to analyse.

## 2. Background

### 2.1 Why investigate variations in pictorial techniques?

Technological studies in rock paintings are useful to understand the technical and economic framework of rock art production as well as to identify possible diachronic trends in these domains (see e.g. [2, 4–6, 8–10, 14, 19, 35, 51–75]). From a technological point of view, two main aspects are expected to vary between paintings: the way the paint was prepared and the way it was applied. Variations in application procedures can reflect functional choices related, for instance, to the intention of producing fine lines or large figures, regular or irregular edges, transparencies or opacities between the line and the painted surface. Paint preparation can be driven by colour choices and functional requirements, such as the ability for the prepared paint mixture to stick to the substrate or to dry rapidly. The pictorial mixture might be more or less viscous, depending on the way it is prepared [4]. In this regard, the choice of the binder is also an important parameter. Technical choices may be limited by raw material availability, depending on a geological context for the inorganic part of the paint mixture. Restrictions in the choice of paint materials may have influenced both preparation and application techniques or, conversely, the choice of one particular technique may have guided the choice of raw materials and the use of other techniques. Finally, binders and pigments are not independent parameters. The choice of one influence the choice of the other. They are also highly influenced by a society's overall technical and symbolic systems (e.g. use of the same binder for different purposes) and the way in which the painter has been taught to mix and apply the paint. In other words, the cultural logic driving the "*chaîne opératoire*" [76] may play a key role in the final appearance of the painting.

Consistency has been observed at El Castillo between these three parameters, choice of raw materials, preparation and application techniques: the paintings that are likely to have been blown with two tubes were made with a fine-grained and pure hematite pictorial mixture, while those thought to have been applied with the mouth or with a single tube contain coarse fragments of pigments and a mixture of iron oxide and clay minerals [4]. This might be due to the fact that the former was more liquid and better suited to the two-tube blowing technique, traditionally known as the airbrush technique. On the contrary, when there are few functional or raw material constraints, variations in pictorial techniques are more likely related to technical choices embedded at different degrees within social and symbolic dimensions.

Technological studies at a site or set of sites are also a powerful tool to understand the chronocultural framework behind rock art production [4, 5, 52, 63, 69]. The different choices listed above can reflect cultural traditions, especially if they are driven by arbitrary, social and/or symbolic considerations. Nonetheless, a technical choice can also relate to individual habits, whatever the influence of functional issues, raw material availability constraints, or arbitrary motivations. Simultaneous variations in pictorial techniques and other parameters, such as

location in the cave, the type of designs or their style, are more likely to reflect significant variations in group habits and cultural traditions [4–6, 14, 28, 72]. Between different regions, differences in pictorial techniques could relate to differences in the properties and composition of the raw materials available in the surrounding environments. They may involve distinct cultural traditions, but also variations within a broad set of shared knowledge, depending on the geological context. At a single site or a cluster of close sites, the geological environment can be considered quite stable, except under particular climatic conditions. Variations in technical choices at a site, if significant and consistent with other parameters, likely reflect moments or phases of production. But they may also reflect abrupt changes in raw material availability caused by intense climatic or geophysical changes (erosion of an entire geological horizon in the surroundings for instance). For this reason, artificial pigments whose date of invention or first import in a region is known are more reliable chronological markers than natural local pigments [69]. The same might be true for paint preparation or application techniques that could be accurately dated at contemporaneous archaeological sites. In addition, paint recipes may vary as a function of changes in trade networks and patterns of cultural exchange. It is also possible that groups with different artistic practices came to paint in the same cave roughly during the same period. Available dating methods do not allow for the chronological precision required to investigate such sources of variation.

## 2.2 How to identify pictorial techniques?

Paint application techniques have often been investigated through *in situ* microscopic examination of paintings, with macrophotography being used [38, 52]. They are now completed by *in situ* microscopic images [12]. They are simple, efficient methods that allow the distribution of the pictorial mixture on the substrate to be observed, and also the particle size of the paint components to be characterised. In some cases, they have been combined with experimental programs [38, 77].

Paint preparation used to be and is still commonly investigated through micro-sampling and micro-analysis of pictorial layers (see e.g. [2, 4, 9, 10, 15, 52–56, 58, 61, 62, 64–67, 70–75, 78–81]). Micro-sampling allows various information on the pictorial layers to be collected, since samples can be observed as such in the laboratory and analysed on cross-sections or powder samples. The grain-size, crystal shape, elemental and mineral composition, along with the spatial arrangement of the paint components, can be identified and compared between different samples. These analyses give information on the raw materials that were used to make the paint and, to a lesser extent, on the way these materials were prepared. This approach has been formalised by the determination of significant compositional groups among the sampled paintings, groups referred to as '*pots de peinture*' ('paint pots'; [57, 58, 82]). Different 'paint pots' might represent different paint recipes, but also the acquisition of different raw materials. The scientific limitation of micro-sampling and micro-analyses is the variable and possibly low representativeness of the samples area. Its obvious main disadvantage is its destructiveness.

Systematic sampling of rock paintings is no longer recommended for the study of rock art. The respect of these paintings is now favoured in curation policy in Europe and elsewhere. The development of transportable *in situ* analytical instruments has been proposed as a means to replace invasive sampling approaches in order to determine pigment composition and significant compositional differences between paints [1, 21, 27, 28, 34, 36]. Two techniques are commonly used: portable Raman spectrometry and portable X-ray fluorescence (pXRF). Raman spectrometry allows the identification of the main minerals responsible for the colour of a paint, *sensu stricto* the pigment [16, 21, 24, 25, 27, 32, 65]. In some cases, mixtures of

minerals composing the pictorial layers have been identified [21]. The main limitations of this method are the strong fluorescence phenomenon induced by the substrate and/or deposits on the pictorial layers and the difficulty of distinguishing the minerals composing the pictorial layers from those belonging to the substrate and alteration deposits.

*In situ* pXRF spectrometry is the most frequently used method to identify pigment composition in rock art (Table 1). It has been successfully applied to rock shelters, cave walls and mobiliary art for the identification of the main elements composing pictorial layers. Theoretically, several families of pigments can be identified with this equipment: iron-based yellow and red pigments, black manganese oxides, white phosphates, carbonates, titanium oxides. For black paintings, the absence of manganese and iron on the spectrum is an indication in favour of the use of pigments rich in carbon. In some cases, the type of pigment or differences in pigment composition were used to support chronological attributions [5, 69, 83]. However, data acquisition, processing and interpretation remain challenging in these contexts because 1) the pictorial layers are systematically thinner than the penetration depth of the X-ray emissions; 2) they cannot be physically separated from overlaying alteration deposits or from the underling substrate; 3) rock walls are very irregular in shape and being in contact with the paintings is not always allowed nor possible; 4) the rock substrate is often heterogeneous in composition and covered by various secondary deposits (see e.g. [12, 19, 20, 27, 36]). In particular, compositional variations detected by *in situ* XRF analyses of red paints are often difficult to distinguish from compositional variations in the substrate and in alteration deposits, likely because the iron signal is too weak (thin pictorial layers), too variable (variations in the thickness of alteration deposits and/or pictorial layers), or because discriminant elements are below detection limits [13, 18, 20, 26, 33]. In several pioneering studies, differences between red paints were observed, but no consistent clustering was detected; only one or two figures could be differentiated within the sample being compared with no clear correlation with other parameters, such as graphic style or theme represented [12, 17, 34–36, 84]. Such random differences in composition may not relate to intentional technical choices. They could be due to undetected variations in the composition of the substrate (presence of inclusions, concretions, etc.), undetected presence of alteration deposits (Mn-enriched deposits for instance) or the presence of repaints [4]. Only one study has found consistent differences between stylistic groups of red paintings [28]. The low number of figures analysed (only 6) makes the results preliminary, although promising.

So far, only one painting experimental program was performed to assess the pertinence of pXRF analyses [14]. An experimental approach reveals to be very useful to evaluate the informative potential of *in situ* pXRF results when this technique is applied to paintings and, if necessary, to design means to improve its reliability.

### 2.3 Issues of terminology and methodology related to *in-situ* analyses

The impossibility of physically separating a pictorial layer from its surrounding context by pXRF reveals terminological limitations in rock painting analyses. Usually, the pictorial layer is distinguished from its substrate and overlaying secondary deposits that naturally form or accumulate on the surface of caves or rock shelters, such as coatings of calcite, soluble salts, soot, etc. [2, 19, 23, 58, 70, 90, 91]. Strictly speaking, the substrate is the bedrock on which the paintings were made, but it can also refer to the surface on which the paintings are applied, when it is not possible to separate the superficial layer of the bedrock from secondary alteration deposits underlying the paint layer. When that superficial layer is covered by secondary deposits, analyses of cross-sections will reveal them, and they will be distinguished during the analysis from the rock substrate [2, 15, 19, 23, 58, 70]. The characterisation of each of these

**Table 1. Details of previous pXRF analyses carried out in rock shelters or caves for the characterisation of rock paintings.**

| Study | Location of the paintings | | | Instrumental parameters | | | | | | | | Sampling | | |
| --- | --- | --- | --- | --- | --- | --- | --- | --- | --- | --- | --- | --- | --- | --- |
| | Name of the site | Type of substrate | Rock substrate | Type of instrument | Anode of X-ray tube | Voltage | Intensity | Counting time | Detector | Filters/ colimaters | Detection of light elements | Figures | Analyses of pigments | Analyses of the substrate and alterations |
| Chanteraud et al. 2021 | Grotte aux Points, France | Cave wall | Limestone | ELIO XGLab | Rh | 40 kV | 100 mA | 300 s | SDD | | From Al (air system) | 32 | 43 | 23 |
| Trosseau et al. 2021 | Font-de-Gaume, France | Cave wall | Limestone | ELIO XG-Lab | Rh | 40 kV | 40 µA | 600 s | SDD | ? | | | 78 (including the substrate) | Yes |
| Huntley et al. 2021 | BH15-01, Central Pilbara, Australia | Rock shelter wall | BIF | Bruker Titan S1 800 | Rh | 45 kV—15 kV | 10.45 µA—31.55 µA | 90 s | SDD | Ti-Al filter—no filter | From Al (air system) | ? | 21 | ? |
| Castañeda et al. 2019 [87] | Eagle Cave and Sayles Adobe, Texas | Pebbles | ? | Innov-X Systems Alpha Series | Ag | 40 kV | ? | 60 s | Si-PIN | ? | No (from Ti to Bi) | 70 | 139 | 115 |
| Lebon et al. 2019 | Doi Pha Kan, Thailand | Rock shelter wall | Limestone | ELIO XGLab | Rh | 40 kV | 100 mA | 300 s | SDD | | From Al (air system) | 6 | ? | Yes |
| Mauran et al. 2019 | 8 sites, Erongo, Namibia | Rock shelter wall, stained artefacts and grinding stones | Granite | ELIO XGLab | Rh | 20 kV | 200 mA | 300 s | SDD | | From Al (air system) | 34 | >57 | 41 |
| Bedford et al. 2018 | Pleito Creek, USA | Rock shelter wall | ? | Bruker Tracer III-V | Rh | 40 kV | 3,4µA | 60 s | Si-PIN | | ? | 1 | 6 | 4 |
| Huntley et al. 2018 | Kabi Kabi Hand Stencil Site, Australia | Rock shelter wall | Sandstone | Olympus 40 kV Delta Environmental | 4 W Ta | 40 kV—15 kV | 100 µA—80 µA | 180 s | Si-PIN | | No (air system) | 3 | 8 | 11 |
| Gay et al. 2016 (2020) | Rouffignac, France; Font-de-Gaume, France; La Garma, Spain | Cave wall | Limestone | Home made—MOXTEK X-ray tube | Pa | 40 kV | - | - | SDD | | From Al (air system) | | | |
| Rifkin et al. 2016 | Apollo 11, Namibia | Plaques | ? | Bruker Tracer III SD | ? | 40 kV-15 kV | 10 µA—55 µA | 120 s | SDD | Al-Ti filter—no filter | From Al (vacuum system) | 6 | 37 | 6 |
| Wallis et al. 2016 [88] | VSTA_20140611_1 quarry | Quarry wall | Shale | Bruker Titan S1 800 | Rh | 45 kV—15 kV | 10.45 µA—31.55 µA | 45 analytical s (effective time of detection) | SDD | Ti-Al filter—no filter | From Al (air system) | 3 | | |
| Gay et al. 2015 | La Garma, Spain | Cave wall | Limestone | Home made—MOXTEK X-ray tube | Pa | 40 kV | | | SDD | | | 10+1 | 57 | Yes |
| MacDonald 2016 | 8 sites, Southern Canadian Shield area | Rock shelter wall | Granite, granitic gneiss | Innov-X Delta Premium | Au | 40 kV—15 kV | 0.1 mA | 120s | SDD | ? | From K | 62 | >62x2 | yes |
| Sepulveda et al. 2015 | 6 sites, Lluta to Camarones coastal valleys, Chili | Rock shelter wall | Igneous ferromagnesian volcanic rock | Bruker Tracer III-SD | Rh | 15 kV | 21 µA | 30 s | SDD | | From Al (air system) | | 19 | 6 |
| Beck et al. 2014 | Rouffignac, France | Cave wall | Limestone | Home made | Cu | 40 kV | 700 µA | - | SDD | polycapillary semi-lens | No (air system) | 15 | 26 | 26 |

*(Continued)*

**Table 1.** (Continued)

| Study | Location of the paintings | | | Instrumental parameters | | | | | | | | Sampling | | |
|---|---|---|---|---|---|---|---|---|---|---|---|---|---|---|
| | Name of the site | Type of substrate | Rock substrate | Type of instrument | Anode of X-ray tube | Voltage | Intensity | Counting time | Detector | Filters/colimaters | Detection of light elements | Figures | Analyses of pigments | Analyses of the substrate and alterations |
| Bedford et al. 2014 | Three Springs, California | Rock shelter wall | ? | Bruker Tracer III | ? | 40 kV | 3,4μA | 60 s | ? | | No (air system) | ? | 150 (including substrate) | Yes |
| Hernanz et al. 2014 | Multiple sites, Spain | Shelter and cave wall | Limestone? Dolomite? | X-MET5100, Oxford instrument | Rh | 45 kV | - | 50 s | SDD | - | No (air system) | - | - | - |
| Koenig et al. 2014 | 10 sites, Lower Pecos Canyonlands, USA | Rock shelter wall | Limestone | Innov-X Systems Alpha Series | Ag | 40 kV | - | 30 s | Si-PIN | - | No (air system) | - | 225 | 23 |
| Lopes-Montalvo et al. 2014 | Cova Remigia rock shelters, Spain | Rock shelter wall | Limestone? | Home made | Ag | 30 kV | 0.1 mA | 200 s | Si-PIN | Al pin-hole | No (air system) | 25 | 34 | 18 |
| Pitarch et al. 2014 | Los Chaparros, Spain | Rock shelter wall | Limestone | X-MET5100, Oxford instrument | Rh | 45 kV | - | 50 s | SDD | - | No (air system) | 2 | 12 | 5+4(black dendrites) |
| Wesley et al. 2014 | 4 sites, Red Lily Lagoon area, Australia | Rock shelter wall | Sandstone | Bruker Tracer III-V | Rh | 40 kV - 15 kV | 15 μA—1.1 μA | 180 s | Si-PIN | Cu-Ti-Al | | 32 | | Yes |
| Loendorf and Loendorf 2013 | Picture Cave, Texas, USA | Rock shelter wall | Limestone | Bruker Tracer III-V | Rh | 40 kV—15 kV | 12 μm | 150 s | Si-PIN | Al-Cu-Ti and Ti | Vacuum | 9 | ? | Yes |
| Olivares et al. 2013 | La Peña Cave, Spain | Cave wall | Limestone | X-MET5100, Oxford instrument | Rh | 45 kV | - | 50 s | SDD | - | Yes (detector?) | | 56 | |
| Roldán et al. 2013 (2016) | Parpalló cave, Spain | Plaquettes | Limestone-sandstone | Home made | Ag | 30 kV | - | - | Si-PIN | Al pinholes | No (air system) | 18 | 40 | 27 |
| Velliky and Reimer 2013 [89] | DjRt–10 and EaRu–9, Squamish Valley, British Colombia, Canada | Rock shelter wall | Granite | Bruker Tracer III-V | Rh | 40 kV—15 kV | 12 μm | 150 s | Si-PIN | Al-Cu-Ti and Ti | Vacuum | 3 | 15 | 15 |
| Huntley 2012 | BR29 Rock Shelter | Rock skelter wall | Sandstone | Bruker Tracer III-V | Rh | 12 kV | 20 μm | 300 s | Si-PIN | Ti 0.0254μm filter | From Al (vacuum system) | | 2 (experiment) +5(rock art) | 27 (experiment) |
| Nuevo et al. 2012 | Abrigo dos Gaivões and Igreja dos Mouros caves | Cave wall | ? | Home made | Ag | 30 kV | 30 μm | | Si-PIN | - | No (air system) | | 10 | 5 |
| Roldán et al. 2010 | Saltadora rock shelters VII, VIII and IX, Spain | Rock shelter wall | Calcareous rock | Home made | Ag | 30kV | 0.1 mA | 200 s | Si-PIN | Al pin-hole | Vacuum | 41 | 57 | 15 |
| Appoloni et al. 2009 [85] | Jaguariaíva 1 rockshelter, Brazil | Rock shelter wall | Sandstone | Home made | Ag | ? | - | 300 s | Si-PIN | Ag | No (air system) | - | 19 (including substrate) | Yes |
| de Sanoit et al. 2005 | Rouffignac, France | Cave wall | Limestone | Home made | - | 15kV | 40 μA | - | SDD | - | No (air system) | 7 | 12 | |
| Newman and Loendorf 2005 [86] | Musselshels site, Montana; Castle Gardens Wyoming | Rock shelter wall | Sandstone | Innov-X systems XT series model 700 | - | - | - | 90 s | - | - | No (air system) | - | 46 | - |

layers by *in situ* pXRF analyses is not possible. The pictorial layer is commonly characterised by comparing the signal of the paintings with the signal of the surface close to them. This is what, depending on the authors, is referred to as the analysis of the substrate, i.e. the cave wall, rock wall, or unpainted rock background, [2, 5, 12, 20, 36, 39]. The signal obtained is a combination of the signal of the bedrock plus the signal of its superficial layer and its alteration deposits. In the present work, a fragment of the cave wall was sampled and its inner part (fresh rock) analysed and compared to its outer part and other locations of the cave wall in order to distinguish the composition of the bedrock from the composition of its surface as exposed in the cave.

More recently, some authors have proposed another method to separate the pXRF signal of the paint components from the signal of their 'environment' [5, 36]. They used the measurement from the cave wall near a painting as a blank and its signal was subtracted from the signal of the painting. The remaining signal was considered to be that of the pictorial layer alone. The terminology introduced in this study is very useful as the term 'environment' is neutral and can be used whatever the substrate is made of and covered with. In the present work, we choose to keep this terminology. However, we will not use cave wall signals as blank signals because we cannot be sure that the environment beneath the paintings is the same as the environment close to them. We will instead evaluate how the cave wall signal varies from place to place, how variations in the environment may affect the signal of the paintings, and how these variations affect the clustering of the measurements.

The last limitation to be taken into account is the distance of the pXRF instrument from the paintings and the irregularities of their surface. If the instrument is not in contact with the paintings, the signal is affected by the layer of air present between the two. This results in a decrease in the intensity and peak ratios of light elements (see e.g. [46]). In some studies, it was possible to position the XRF instrument in contact with the paintings (see e.g. [19, 69, 83, 84, 92]). In our case, however, as in most similar studies in Europe, we were not allowed to touch the wall and, although always in the range of few millimetres, the precise distance from the pictorial layer could not be controlled. The irregular shape of the wall was a supplementary limitation for close contacts. As a consequence, peak ratios between light and heavy elements may vary from one spectrum to another. Surface irregularities result in similar problems [93, 94]. In order to evaluate the impact of these factors on the results, we carefully analysed the spectra, and as in previous studies, we applied statistical analyses to the spectra [13, 25, 95]. This approach allowed us to evaluate the potential factors affecting the signals.

## 2.4 Archaeological context

El Castillo cave opens onto the side of a mountain of the same name near the village of Puente Viesgo (Cantabria, Spain) (Fig 1). Featuring rich evidence of human occupation since at least 150,000 years ago, El Castillo and the nearby caves contain remarkable stratigraphic sequences and art, whose discovery and study has played a key role in Palaeolithic archaeology. El Castillo was discovered in 1903. Excavations at the entrance exposed an 18–20 m deep stratigraphy comprising 26 alternating sterile and anthropogenic layers [96]. From top to bottom, the archaeological sequence begins with Medieval and Chalcolithic deposits overlying Azilian and Magdalenian levels [96, 97]. Solutrean, Gravettian and Aurignacian occupations overlie layer 18, attributed to the Late Mousterian [98] or Initial Aurignacian [97], under which we find levels recording the regional evolution of the Mousterian. The basal deposit contains Acheulean lithic artefacts. The karstic network beyond the entrance contains one of the most conspicuous Palaeolithic art ensembles of Western Europe, with numerous engravings, drawings and paintings covering the full range of the themes, techniques and styles of the period [40–42, 45, 99].

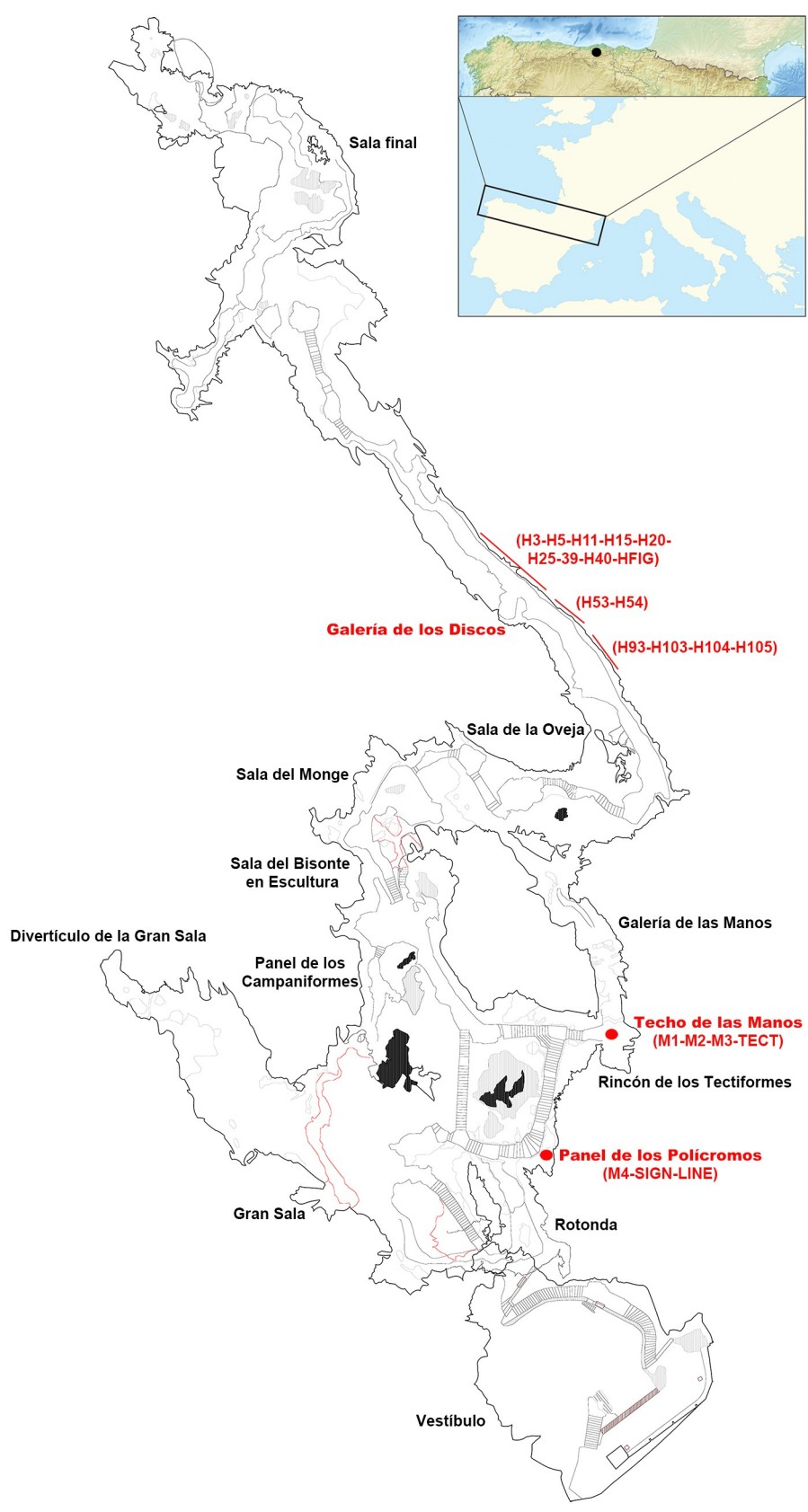

**Fig 1. Location of El Castillo site and map of the cave with the location of the different panels (source of the map: open source https://www.naturalearthdata.com).**

The paintings are distributed around various parts of the cave and organised in different panels. Some panels are polychromic, other entirely made with red paints. The latter largely dominate when all the abstract designs, such as disks and lines, are included.

## 3. Materials and method

### 3.1 El Castillo paintings

A total of 14 red disks, four red hand stencils, one yellow bison, one red tectiform, one red vulvar sign, one red digital line and one red lineal composition were analysed (Fig 2; Table 2). We selected red disks from different parts of the Galería de Los Discos: part 1, end of the corridor; part 2, middle of the corridor; and part 3, beginning of the corridor. Since access to the red disks that have been sampled and analysed in a previous study was impracticable with the pXRF instrument [4], we analysed similar disks located in the same areas. The dated disks [44] were not selected for analysis. The first one (date O-69-80) because it is located in a niche that is not accessible with the pXRF instrument and the other (O-83) because it is overlaid by a thick layer of calcite.

The red disks from the Galería de los Discos were compared with a figure from part 1 of the Galería, three hand stencils from the Techo de las Manos, one from the Panel de los Polícromos, two figures (bison and tectiform) from the Techo de las Manos and a line and vulvar sign from the Panel de los Polícromos (Fig 2; Table 2). The dated hand stencils and disks are older than 34 ka and may belong to the Aurignacian or, possibly, the Mousterian [44]. The selected figures are considered to be younger than the disks and hand stencils, based on the stratigraphy of the paintings and their style [44, 100]. The style of the yellow bison suggests a Gravettian age, while the red tectiform and vulvar sign are attributed to the Magdalenian.

A total of 13 pXRF measurements were taken on the substrate close to the paintings. Only the flattest surfaces were analysed. Measurements of the substrate between two paintings was used as a reference for both. That is why the number of substrate measurements is lower than the number of paintings analysed. In the Panel de los Polícromos, a red natural deposit (Wall red) was also measured. The exact location of pXRF measurements is given in (S1-S6 Figs in S1 File).

### 3.2 Experiments

**3.2.1 First experiment (Exp 1).** The aim of the first experiment we carried out was to determine the extent to which red pigments with different compositions could be distinguished with no control on the distance of the pXRF equipment from the wall. As a substrate, we used a large block of Rupelian marine limestone from the quarry of Lugasson, Gironde, France. Experimental paint was applied on an irregular fresh fracture of the block. The absence of concretions and alteration deposits considerably decreased the compositional variations of the substrate. Two colouring materials were used:

- S: Ref-Fe1: ferruginous shale from De Hoop, South Africa (home-made reference, see [101, 102]).

- H: Blutstein, Kremer pigment, Blutstein Type 2004.

Their elemental composition is given in Table 3. These two reddish pigments were mixed with water in different proportions and applied on the substrate in different ways (Fig 3A). For the present study, we kept only preparation type 1 (H1 and S1), which resulted in the thinner

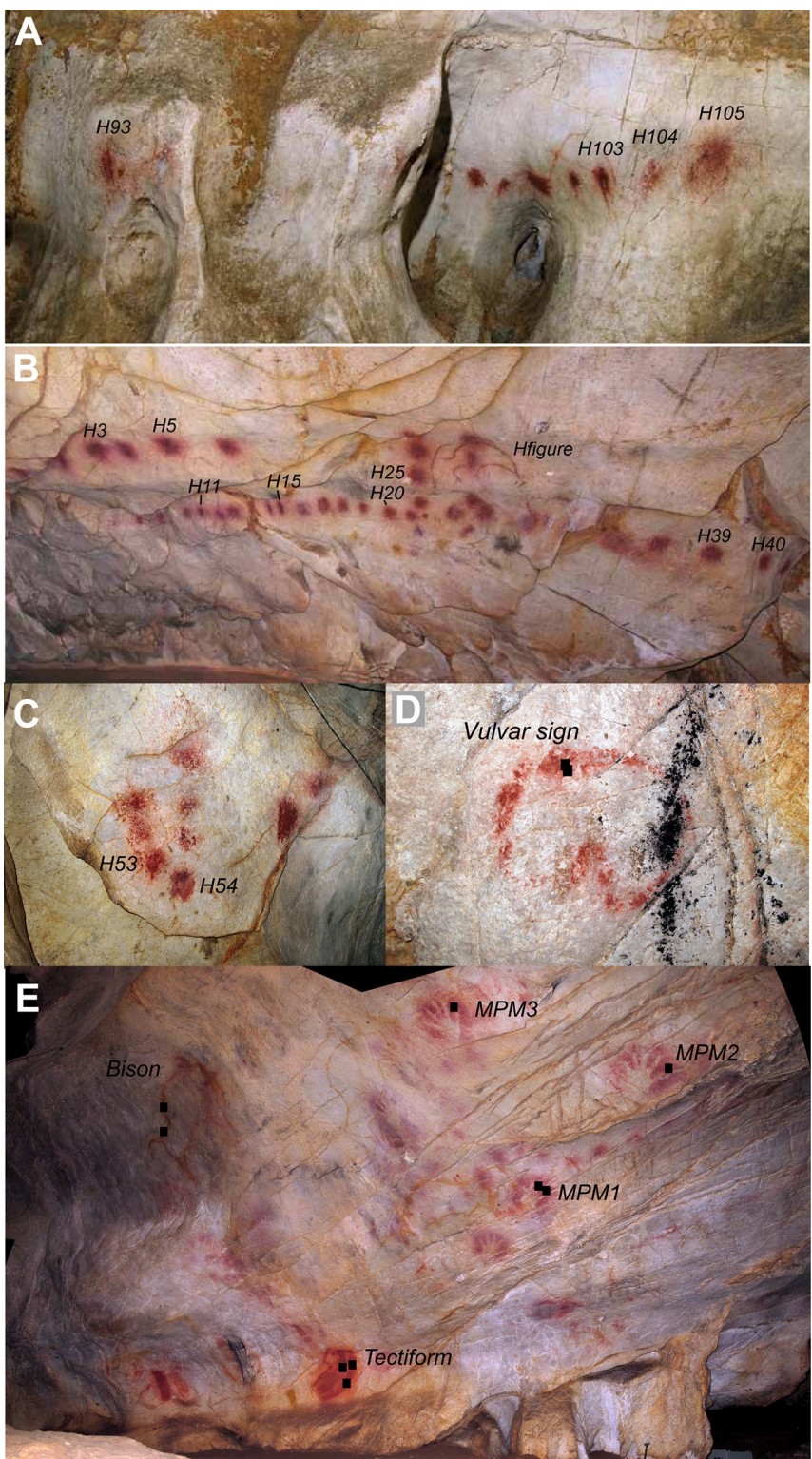

**Fig 2. Pictures of the main paintings analysed in the present study.** Black squares indicate where pXRF analyses were performed. A: Galería de los Discos, beginning (H-1). B: Galería de los Discos, end (H-3). C: Galería de los Discos, middle (H-2). D: Panel de los Polícromos, Vulvar sign. E: Techo de las Manos.

**Table 2. List of the paintings analysed, type of analyses performed and number of pXRF spectra used in the present study.**

| PANEL | REPRESENTATION | REF. PAINTING | pXRF analyses | Microscope | Micro-sampling (d'Errico et al. 2016) |
|---|---|---|---|---|---|
| *Galería Final* | *DISK* | *LL1* | - | - | *X* |
| *Galería de los Discos, H-end* | *DISK* | *H3* | *3* | - | - |
| | *DISK* | *H5* | *1* | - | - |
| | *DISK* | *H11* | *2* | - | - |
| | *DISK* | *H15* | *3* | - | - |
| | *DISK* | *H14* | - | - | *X* |
| | *DISK* | *H20* | *3* | - | - |
| | *DISK* | *H25* | *2* | - | - |
| | *LINEAR COMPOSITION* | *HFIG* | *2* | - | - |
| | *DISK* | *H39* | *3* | - | - |
| | *DISK* | *H40* | *3* | - | - |
| *Galería de los Discos, H-middle* | *DISK* | *H53* | *2* | - | - |
| | *DISK* | *H54* | *3* | - | - |
| | *DISK* | *H58* | - | - | *X* |
| *Galería de los Discos, H-beginning* | *DISK* | *H93* | *3* | - | - |
| | *DISK* | *H99* | - | - | *X* |
| | *DISK* | *H103* | *1* | *X* | - |
| | *DISK* | *H104* | *2* | - | - |
| | *DISK* | *H105* | *2* | *X* | - |
| *Techo de los Manos* | *BISON* | *BIS* | *2* | - | - |
| | *HAND STENCIL* | *M1* | *2* | - | - |
| | *HAND STENCIL* | *M2* | *2* | - | - |
| | *HAND STENCIL* | *M3* | *1* | - | - |
| | *TECTIFORM* | *TECT* | *3* | - | - |
| *Sala de los Polychromos* | *HAND STENCIL* | *M4* | *2* | - | - |
| | *VULVAR SIGN* | *SIGN* | *2* | - | - |
| | *DIGITAL LINE* | *LINE* | *4* | - | - |
| *TOTAL PAINTINGS* | | | *53* | - | - |
| *TOTAL CAVE WALL* | | | *13* | - | - |
| *TOTAL* | | | *66* | *2* | *4* |

**Table 3. Elemental composition (major elements) of the two pigments used in the experiments.**

| Ref. | Name | Fe$_2$O$_3$ | SiO$_2$ | Al$_2$O$_3$ | CaO | MgO | K$_2$O | TiO$_2$ | P$_2$O$_5$ | MnO | Na$_2$O | SO$_3$ | CO$_2$ | H$_2$O (struct.) | PF* |
|---|---|---|---|---|---|---|---|---|---|---|---|---|---|---|---|
| S | Ref.-Fe1 | **8.152** | **47.34** | 26.417 | **0.031** | 0.989 | **5.474** | **1.507** | 0.04 | 0.0118 | 1.101 | - | - | - | 8.84 |
| H | Blutstein | **80** | **7.8** | 3.2 | **2.1** | 1.5 | **1** | <**0.1** | <0.1 | <0.1 | <0.05 | <0.05 | 3.4 | (strukt.) | - |

*PF: ignition loss (perte au feu).

For Blutstein Type 2004, the data are those given by Kremer pigment manufacture. For Ref-Fe1, see Dayet et al. 2014; 2019. Bold values: elemental contents that could theoritically be used fo differentiate the two colouring materials with the instrumental conditions used.

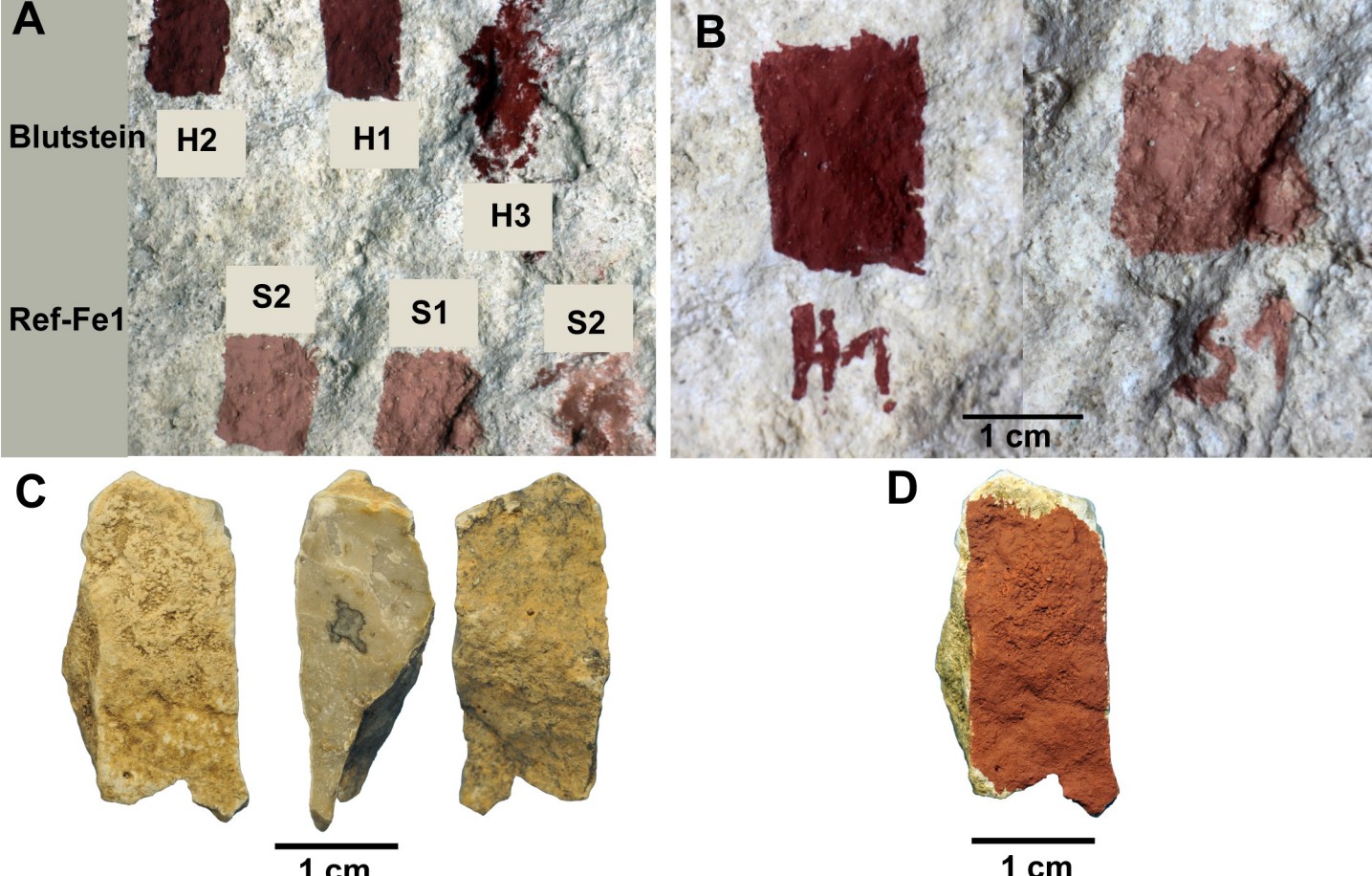

**Fig 3. Pictures of the experimental paintings.** A: paintings made on the Rupelian limestone (1: mixed with water, applied with a brush; 2: mixed with less water, applied with a brush; 3: powder directly applied with a finger). B: pictures of the selected experimental paintings H1 and S1 on the Rupelian limestone; C: fragment of El Castillo cave wall. D: same fragment with painting H1 (Exp2).

and more homogenous paint layer (Fig 3B). The paints, made by mixing 0.3g of pigment in 1.5mL of water, were applied with a brush.

**3.2.2 Second experiment (Exp 2).** The purpose of the second experiment was to create a reference painting comparable to the El Castillo paintings in order to help the interpretation of the archaeological results and evaluate the extent to which differences in provenience or paints' pot could be distinguished when working at El Castillo. We applied a paint on a fragment of El Castillo limestone with the H1 pigment mixture (Blutstein+water; Fig 3C and 3D). The Blutstein pigment is rich in Fe (about 80%) and it is also composed of a little Si and Al, a composition that corresponds to the composition of the El Castillo red disks [4]. The provenience of the two pigments is, however, clearly different. The paint was applied with a brush in order to produce a homogenous paint layer.

### 3.3 Methods

**3.3.1 Digital microscope.** For the *in situ* microscopic observations, we used a Hirox VCR 800 digital microscope attached to the end of a metal arm mounted on a heavy tripod and driven by a crank handle in order to avoid vibrations, allow precise movement of the

microscope and avoid accidental contact with the cave wall. Observations were conducted at magnification ranging between 20x and 160x.

**3.3.2 Portable X-ray spectrometer (pXRF).** The pXRF measurements were carried out using a portable SPECTRO xSORT X-ray fluorescence spectrometer (Ametek), equipped with a silicon drift detector (SDD) and a low-power W X-ray tube with an excitation source of 40kV. Measurements were acquired in the air without contact with the paintings, by fixing the device on a dedicated metal stand attached to the same equipment described above for the digital microscope. The working distance varied according to the flatness of the painting surface, but was kept below 3mm. An area of 8mm in diameter was analysed. Spectra acquisition time was set to 300s. The spectrometer was internally calibrated by an automated measurement of the contents of a standard metal shutter. We took at least two pXRF measurements per painting and up to four when possible. Only the spectra showing the highest Fe peaks were kept because the lowest Fe peaks were observed when secondary deposits on the paintings were the thickest. The number of spectra per painting used in data processing is given in Table 2. Measurements were also performed on the experimental paints, first on the substrate before the application of the paintings, and then on the paintings themselves.

## 3.4 pXRF data processing

**3.4.1 Processing of pXRF spectra.** As discussed above, *in situ* pXRF analyses have limitations. We chose to compare the raw spectra first, in order to get an idea of how the measurement conditions impacted them. As a second step, the spectra were normalised by the total counts per spectrum, using the same range of energy (see [17, 18]). We also tested a normalisation method inspired by the Fe-ratios normalisation used in ochre provenance studies [101, 103–108]. The method reduces the differences in trace element composition between ochre samples of varying Fe content. Palaeolithic red and yellow paints are usually composed of iron-rich pigments. Using Fe-ratios may decrease the compositional differences between pictorial layers that have different thicknesses or that are covered by heterogeneous deposits. The following normalisation formula was applied:

$$I_{norm} = \log(I_x - bg_{min})/(A_{Fe} - A_{bgFe})$$

$I_x$ = number of counts for the energy channel considered.

$bg_{min}$ = minimum of the background (number of counts) depending on the range of energy (1.7–4.3keV; 4.3–7.6keV;10.3–10.9keV).

$A_{Fe}$ = area of K-alpha Fe peak, sum of the counts from E = 6.11 to 6.65keV.

$A_{bgFe}$ = approximate background under K-alpha Fe peak, $bg_{min}$ multiplied by the number of channels used to calculate $A_{Fe}$.

Logarithms were preferred because data calculated using ratios did not show a normal distribution.

**3.4.2 Calculation of peak areas.** In order to assess whether the characterisation of iron-rich pictorial layers can be improved by pXRF data processing, we also used the areas of the peaks (netto counts) calculated using the fundamental parameters (direct values calculated by the PDA of the SPECTRO xSORT X-ray instrument). Considering the large differences between elements, we used the centred log ratio (clr) and the logarithm weighted by Fe netto counts (alr) [109]. We used Fe-ratios of netto counts for the same reason we used the Fe-ratios of the spectra. The calculations were performed using CODAPAK software [110]. When more than 25% of the values were missing (netto count = 0), the element was not taken into account. Missing values were replaced by 0.65 multiplied by the minimum peak area detected for the element concerned following the software recommendation.

**3.4.3 Multivariate statistical data processing.**  Statistical analyses were used in order to improve sample comparison. We used Principal Component Analyses (PCA) in order to describe composition variability among the measurements. The variables and individuals retained in each PCA are summarised in Table 4. For the spectra, PCAs were carried out using a selected range of energies: 1.70–10.92 KeV (variables). Energies below 1.70 keV were not used because the instrumental conditions were not suitable for the detection of light elements and energies above 10.92 keV, as the peaks of heavy elements are too weak or absent. As the logarithm of null values cannot be calculated, this means that when the background reaches its minimum, there is a missing value in the Fe-ratio log data. All energies showing missing values were removed from PCA4, as were energy ranges between 7.61 and 10.39 keV because of the high contribution of the X-ray source in this part of the spectra.

Except for PCA6 and PCAc, we used all the measurements performed at El Castillo associated with the second experimental reference (Exp2) as individuals. Cave wall measurements were kept in all PCAs to avoid underestimating the contribution of the cave wall to the results. PCA coordinates were calculated using the FactoMineR library in R software [111], and graphs were done with the factoextra library.

# 4. Results

## 4.1 Microscopic examination

Macroscopic and microscopic examination of the two analysed disks revealed that their external part is composed of pigment drops (Fig 4A to 4C). This corresponds to an application by the blowing technique. Pigment trails were observed when fresh paint was reworked with the fingers (H103; Fig 4B and 4D). The pigment layer of disk H103 is thick and well preserved (Fig 4B and 4F). It is composed of a homogenous dark red fine-grained matrix. Small dark particles and coarse transparent inclusions were detected in the paint. The pigment layer of the disk H105 is very thin (Fig 4A and 4E). It is composed of a dark red fine-grained matrix. No inclusions were observed.

## 4.2 pXRF analyses

**4.2.1 First experiment.**  Paints H1 and S1 can be clearly distinguished from each other by their pXRF spectra (Fig 5A). Paint S1 is characterised by higher Si, K and Ti K-alpha peaks. These peaks are very weak on the spectrum of Rupelian limestone, indicating that they correspond to the pictorial layer. This is consistent with the higher content of these three elements

**Table 4. Summary of the variables (energies and elements) and individuals (measurements) used in the different PCAs.**

| PCA | Variables | Individuals |
|---|---|---|
| **PCA1** | Raw spectra, energies from 1,70 to 10.92 | All *in situ* measurements, Exp 2 |
| **PCA2** | Normalized spectra, energies from 1,70 to 10.92 | All *in situ* measurements, Exp 2 |
| **PCA4** | log of Fe-ratios, energies from 1,70 to 7,60; from 10,31 to 10.92; variables with missing values removed | All *in situ* measurements, Exp 2 |
| **PCAa** | clr Netto counts, elements with less than 25% of null values | All *in situ* measurements, Exp 2 |
| **PCAb** | alr-Fe Netto counts, elements with less than 25% of null values | All *in situ* measurements, Exp 2 |
| **PCAc** | alr-Fe Netto counts, elements with less than 25% of null values | Reddish cave wall and Exp 2 excluded |

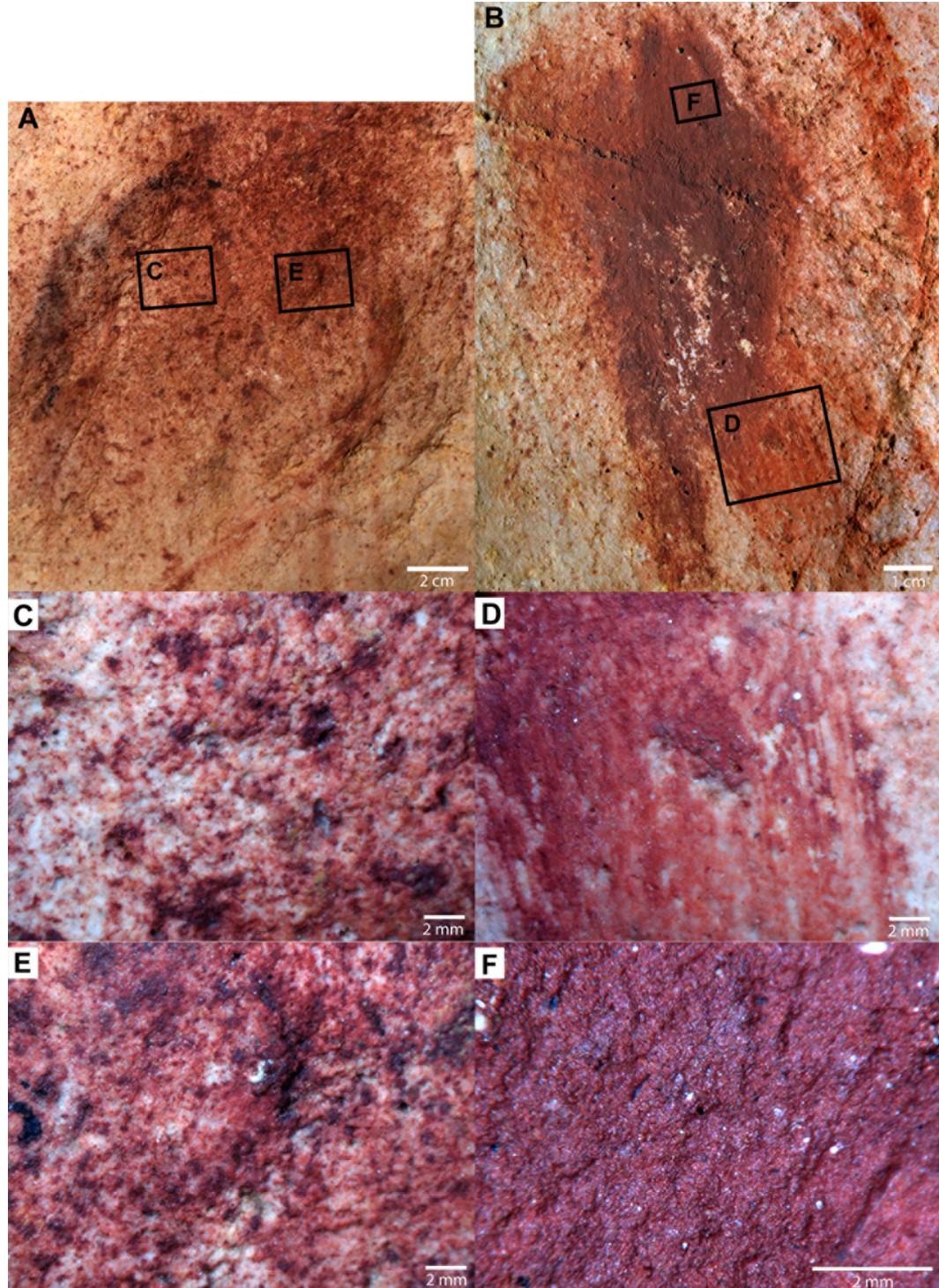

**Fig 4.** Macro and microphotographs of the paintings H105 (A, C, E) and H103 (B, D, F). Colours are not calibrated, which explains the differences in colour between the macro and microphotographs. The light conditions were identical.

in S1 (Table 3). The Fe peaks is higher in paint H1. This is consistent with its higher Fe content. The Ca K peaks are very similar, while the Ar K-alpha peak (present in the air) is a little higher in S1. This pattern indicates a possible difference in air thickness between the two

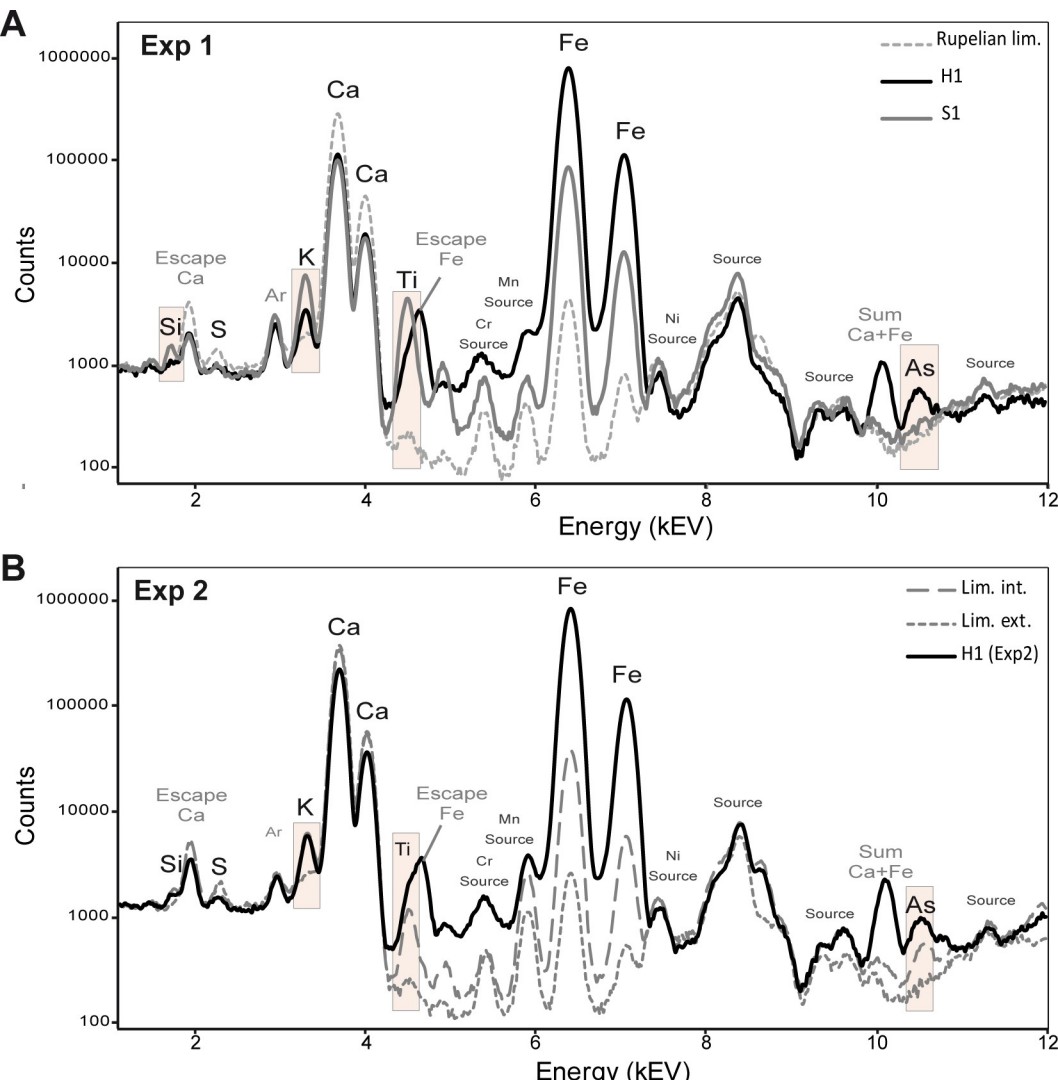

**Fig 5.** Normalised pXRF spectra of the first experiment (A) and the second experiment (Exp2, B); Normalisation by total counts.

measurements. The distance from the substrate was probably higher for S1. In these conditions, the peak heights of the light elements are probably underestimated on the S1 spectrum. In spite of this underestimation, the difference in composition between H1 and S1 remains perfectly detectable.

**4.2.2 Second experiment (Exp2).** The El Castillo limestone used in the second experiment is heterogeneous in composition. Its inner part is composed of higher proportions of Si, K, Ti, Fe and As than the Rupelian limestone (Fig 5B). These elements are, however, detected in lower proportions in its external altered part. The pXRF measurements of the paint made on the external part is characterized by higher K, Ti, Fe and As K-alpha peaks than this latter. The difference in Fe peaks' intensity is the main difference we can observe between the limestone as a whole and the paint layer. These results show that the composition of the Fe-rich pictorial layers can be distinguished from the composition of El Castillo substrate, and that this difference is higher when the environment is depleted in elements such as K, Ti and As.

**4.2.3 El Castillo normalised spectra.** According to the pXRF analyses, all the red paints are likely composed of Fe-based pigments (Fig 6A; Table 5). The yellow paint depicting the figure of a bison, is also composed of a Fe-bearing pigment (Fig 6A). The difference in Fe peak

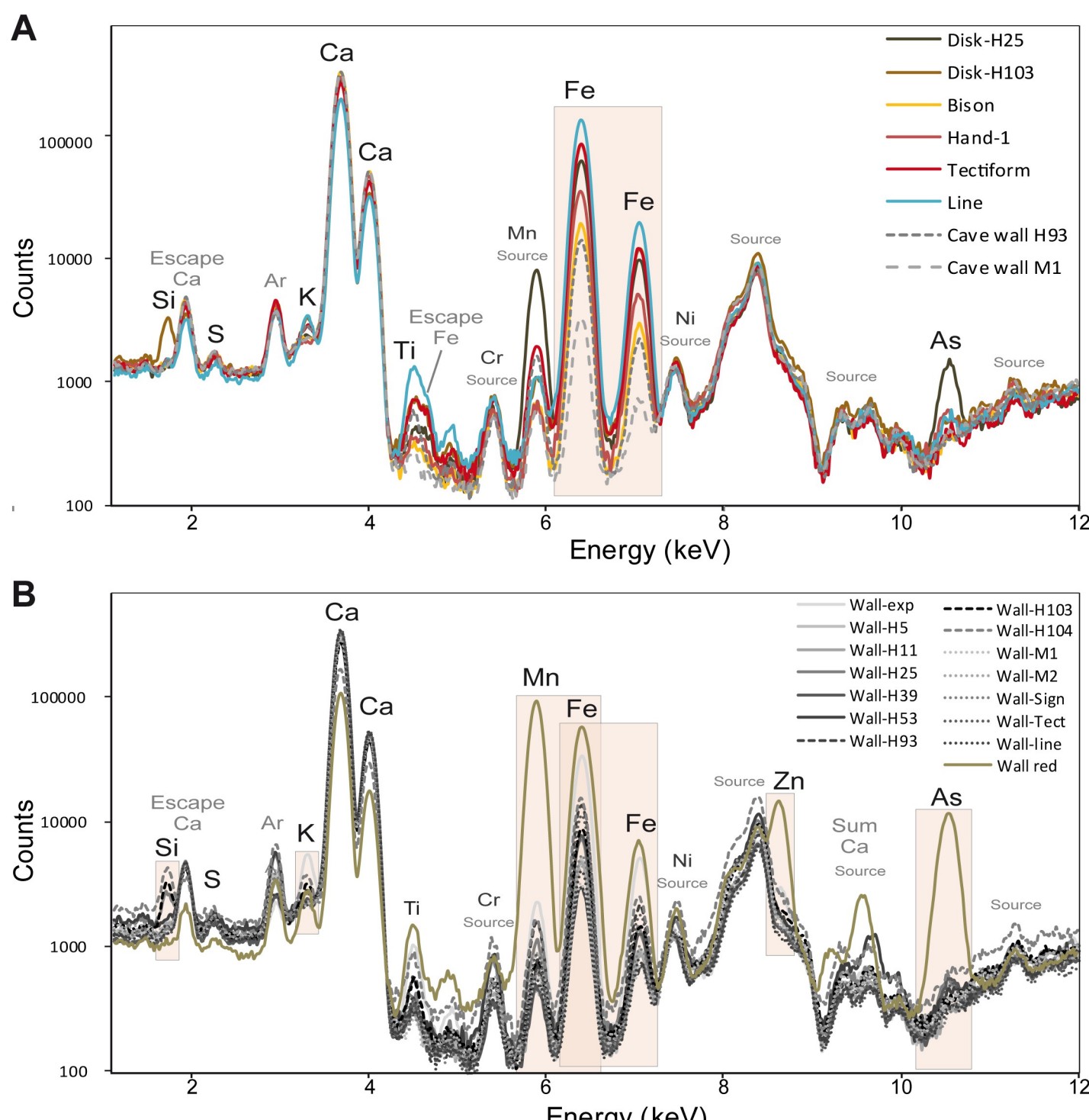

**Fig 6.** Normalised pXRF spectra of a selection of figures with the substrate (A) and normalised pXRF spectra of all the measurements carried out on the cave wall (B).

**Table 5. Netto counts of elements detected by pXRF, calculated with the fundamental parameters.**

| Painting | Type | Si | P | S | K | Ca | Ti | Cr | Mn | Fe | Ni | As | Sr | Y | Total |
|---|---|---|---|---|---|---|---|---|---|---|---|---|---|---|---|
| Exp2_1 | Exp | 11414 | 31206 | 12281 | 116367 | 4561897 | 20086 | 524 | 40138 | 19859546 | 20112 | 17506 | 45304 | 11413 | 24747794 |
| Exp2_2 | Exp | 10139 | 26581 | 10790 | 108546 | 4309299 | 19137 | 525 | 29565 | 20350628 | 18346 | 17711 | 45776 | 15312 | 24962355 |
| H3_1 | Disk | 1407 | 20338 | 5545 | 21582 | 2604060 | 1005 | 578 | 5019 | 213496 | 9223 | nd | 14679 | 4641 | 2901573 |
| H3_2 | Disk | 1453 | 19178 | 5247 | 18558 | 2302612 | 149 | 572 | 3208 | 207113 | 8593 | nd | 9705 | 2407 | 2578795 |
| H3_3 | Disk | 4173 | 40683 | 10512 | 38163 | 4809867 | nd | 576 | 4599 | 382034 | 11180 | nd | 20614 | 732 | 5323133 |
| H5_1 | Disk | 3145 | 19331 | 5517 | 20763 | 2313247 | 1062 | 569 | 7213 | 603833 | 9258 | 1238 | 12500 | 6922 | 3004598 |
| H11_1 | Disk | 3869 | 30926 | 8287 | 30885 | 3795766 | nd | 575 | 4614 | 569420 | 9520 | nd | 16246 | 3839 | 4473947 |
| H11_2 | Disk | 6253 | 39818 | 10583 | 40110 | 4645304 | 1409 | 572 | 13362 | 586275 | 11408 | nd | 28088 | 3659 | 5386841 |
| H15_1 | Disk | 2166 | 15469 | 4354 | 18057 | 1903752 | 435 | 567 | 39005 | 650699 | 7983 | 1145 | 8584 | 1349 | 2653565 |
| H15_2 | Disk | 2706 | 25646 | 7142 | 28114 | 3160429 | 259 | 573 | 97242 | 811192 | 8566 | 3242 | 16403 | 3712 | 4165226 |
| H15_3 | Disk | 2516 | 15150 | 4652 | 18419 | 1837420 | 824 | 566 | 61147 | 590116 | 8902 | 3615 | 12264 | 3314 | 2558905 |
| H20_1 | Disk | 6597 | 27609 | 7891 | 33305 | 3320362 | 3531 | 575 | 8510 | 2052050 | 9286 | 2315 | 18325 | 4313 | 5494669 |
| H20_2 | Disk | 12954 | 26093 | 7583 | 30254 | 3137303 | 5262 | 572 | 8332 | 487582 | 9538 | 1023 | 20876 | 3704 | 3751076 |
| H20_3 | Disk | 6249 | 29035 | 7921 | 34324 | 3400391 | 4394 | 575 | 7380 | 1193541 | 10248 | 1421 | 19648 | 2325 | 4717452 |
| H25_1 | Disk | 2250 | 17335 | 4926 | 28871 | 2191556 | 3675 | 594 | 1581 | 3107547 | 9044 | 3355 | 10228 | 917 | 5381879 |
| H25_2 | Disk | 5063 | 22050 | 6848 | 34916 | 2551864 | 11074 | 584 | 8900 | 1909613 | 10771 | 4592 | 20784 | 3066 | 4590125 |
| H39_1 | Disk | 7443 | 25634 | 6355 | 31068 | 3109413 | 2064 | 572 | 4841 | 1195462 | 8944 | 1197 | 26207 | 6092 | 4425292 |
| H39_2 | Disk | 12236 | 52406 | 14349 | 57152 | 6189905 | 3728 | 565 | 11173 | 1040764 | 12690 | 731 | 34287 | 910 | 7430896 |
| H39_3 | Disk | 9101 | 49941 | 13320 | 61919 | 5689740 | 6529 | 567 | 12269 | 2670121 | 12285 | 3037 | 29967 | 6846 | 8565642 |
| H40_1 | Disk | 10412 | 37710 | 10911 | 61415 | 4384986 | 12386 | 570 | 13823 | 4768707 | 11023 | 645 | 52033 | 3609 | 9368230 |
| H40_2 | Disk | 7343 | 23676 | 5914 | 45264 | 2839420 | 9567 | 580 | 5578 | 5096387 | 9968 | 1659 | 25399 | 2347 | 8073102 |
| H40_3 | Disk | 5058 | 19269 | 5441 | 31977 | 2369365 | 5980 | 580 | 3203 | 3006075 | 8374 | nd | 22534 | 1458 | 5479314 |
| Hfig_1 | Figure | 2291 | 12597 | 2686 | 15168 | 1408994 | 1307 | 561 | 4049 | 149666 | 8119 | 290 | 9436 | 1992 | 1617156 |
| Hfig_2 | Figure | 4068 | 23292 | 6736 | 24092 | 2667276 | 1897 | 568 | 3986 | 207456 | 9358 | nd | 19784 | 3160 | 2971673 |
| H53_1 | Disk | 2752 | 15405 | 4110 | 19424 | 1827476 | 1296 | 572 | 2951 | 1311953 | 7948 | nd | 27402 | 5047 | 3226336 |
| H53_2 | Disk | 2883 | 15646 | 4564 | 23823 | 2001193 | 2070 | 577 | 2774 | 2551964 | 9388 | nd | 25703 | 916 | 4641501 |
| H54_1 | Disk | 3987 | 32883 | 8106 | 33166 | 3922413 | 1019 | 576 | 16904 | 593587 | 10482 | nd | 37014 | 2987 | 4663124 |
| H54_2 | Disk | 8924 | 45029 | 11957 | 48871 | 5429052 | 2996 | 568 | 8119 | 1195999 | 12787 | nd | 64158 | 2463 | 6830923 |
| H54_3 | Disk | 4236 | 20903 | 5482 | 22080 | 2533117 | 684 | 572 | 3343 | 836935 | 9223 | nd | 29350 | 6827 | 3472752 |
| H93_1 | Disk | 7306 | 55063 | 14287 | 58598 | 6444580 | 2978 | 574 | 20368 | 1138474 | 13176 | nd | 37431 | 4246 | 7797081 |
| H93_2 | Disk | 5076 | 27885 | 7520 | 38198 | 3303969 | 5043 | 583 | 5781 | 2508381 | 10014 | nd | 22422 | 5022 | 5939894 |
| H93_3 | Disk | 6034 | 41757 | 10479 | 45015 | 4694889 | 3579 | 574 | 11299 | 532241 | 10951 | nd | 29961 | 2415 | 5389194 |
| H103_1 | Disk | 16739 | 15961 | 4176 | 28596 | 1860991 | 3026 | 558 | 5811 | 921510 | 9566 | nd | 30613 | 6022 | 2903569 |
| H104_1 | Disk | 5358 | 9331 | 2531 | 16943 | 1210022 | 1279 | 448 | 3264 | 215116 | 7502 | nd | 14355 | 879 | 1487028 |
| H104_2 | Disk | 5016 | 5684 | 1721 | 11044 | 613214 | 1622 | 442 | 3108 | 304053 | 7505 | nd | 8468 | 1157 | 963034 |
| H105_1 | Disk | 10835 | 9390 | 2653 | 31034 | 1171118 | 5228 | 568 | 3860 | 1822781 | 10029 | 564 | 21081 | 1606 | 3090747 |
| H105_2 | Disk | 19596 | 16487 | 4944 | 46990 | 1889603 | 7372 | 565 | 7882 | 2138106 | 10781 | nd | 34521 | 1768 | 4178615 |
| Bis_1 | Figure | 4794 | 47701 | 12457 | 43118 | 5797214 | 1318 | 574 | 6866 | 259864 | 11645 | nd | 44518 | 6536 | 6236605 |
| Bis_2 | Figure | 3624 | 32784 | 8812 | 29388 | 3915603 | nd | 572 | 4906 | 257948 | 9565 | nd | 28834 | 5063 | 4297099 |
| M1_1 | Figure | 2236 | 28846 | 7093 | 25733 | 3477022 | nd | 577 | 3736 | 477754 | 9681 | 1704 | 13086 | 2873 | 4050341 |
| M1_3 | Figure | 2547 | 30093 | 7487 | 28746 | 3467136 | 443 | 566 | 7717 | 112568 | 10613 | nd | 19003 | 2740 | 3689659 |
| M2_1 | Figure | 2938 | 28232 | 7888 | 25205 | 3399824 | nd | 566 | 4045 | 83582 | 10645 | nd | 24184 | 2603 | 3589712 |
| M2_2 | Figure | 3132 | 33278 | 8791 | 29980 | 4033382 | nd | 566 | 3557 | 128200 | 9208 | nd | 29703 | 3777 | 4283574 |
| M3_1 | Figure | 1856 | 18804 | 5071 | 17241 | 2250116 | nd | 563 | 4229 | 93316 | 8465 | nd | 15349 | 2413 | 2417423 |
| Tect_2 | Figure | 3107 | 26277 | 6597 | 26884 | 3213791 | 288 | 579 | 4236 | 334082 | 9299 | nd | 34806 | 4955 | 3664901 |
| Tect_3 | Figure | 3396 | 24577 | 6886 | 27646 | 2951808 | 1248 | 576 | 6659 | 758405 | 8402 | 495 | 18784 | 2527 | 3811409 |
| Tect_4 | Figure | 3893 | 22166 | 5159 | 29471 | 2672354 | 3621 | 573 | 17113 | 994284 | 9485 | nd | 12826 | 1721 | 3772666 |

*(Continued)*

**Table 5.** (Continued)

| Painting | Type | Si | P | S | K | Ca | Ti | Cr | Mn | Fe | Ni | As | Sr | Y | Total |
|----------|------|-----|-----|-----|-----|-------|-----|-----|-----|-----|-----|-----|-----|-----|-------|
| M4_1 | Figure | 6735 | 60605 | 16026 | 51491 | 7325295 | nd | 572 | 6980 | 408303 | 13959 | nd | 61977 | 11479 | 7963422 |
| M4_2 | Figure | 1438 | 12097 | 3476 | 11999 | 1405459 | nd | 567 | 3684 | 439688 | 9005 | 1430 | 17878 | 4226 | 1910947 |
| Line_1 | Line | 9440 | 37864 | 10381 | 60789 | 4384686 | 13986 | 570 | 12212 | 1173130 | 12189 | nd | 48148 | 4086 | 5767481 |
| Line_2 | Line | 4754 | 24841 | 6222 | 37937 | 2935625 | 8773 | 577 | 7264 | 1421399 | 9570 | nd | 26643 | 873 | 4484478 |
| Line_3 | Line | 2202 | 10815 | 2790 | 16789 | 1267673 | 3477 | 352 | 3210 | 504753 | 5486 | nd | 14996 | 4415 | 1836958 |
| Line_4 | Line | 5222 | 20078 | 5758 | 39961 | 2390878 | 12509 | 576 | 7449 | 1878705 | 11161 | nd | 29145 | 4240 | 4405682 |
| Sign_1 | Figure | 2454 | 25194 | 6924 | 22884 | 3032249 | nd | 570 | 4094 | 101390 | 9391 | nd | 27760 | 7451 | 3240361 |
| Sign_2 | Figure | 3128 | 32410 | 8247 | 29439 | 3851713 | 125 | 570 | 5332 | 150773 | 9256 | nd | 36135 | 7737 | 4134865 |
| Wall Exp2 | Wall | 6813 | 63316 | 17521 | 56650 | 7681095 | nd | 560 | 22495 | 42966 | 17519 | nd | 59004 | 3421 | 7971360 |
| Wall H5 | Wall | 3993 | 34456 | 10032 | 30924 | 4111481 | 161 | 566 | 3640 | 69456 | 9102 | nd | 21588 | 4047 | 4299446 |
| Wall H11 | Wall | 7345 | 46727 | 12418 | 42161 | 5425281 | nd | 565 | 4038 | 85717 | 10932 | nd | 40233 | 4069 | 5679486 |
| Wall H25 | Wall | 10816 | 21097 | 4980 | 19612 | 2436027 | 344 | 568 | 6739 | 70592 | 9229 | nd | 14079 | 956 | 2595039 |
| Wall H39 | Wall | 9979 | 62280 | 16403 | 59268 | 7484469 | 1992 | 561 | 6129 | 194413 | 13128 | nd | 48649 | 1169 | 7898440 |
| Wall H53 | Wall | 1962 | 16971 | 4150 | 15441 | 2060118 | nd | 559 | 1795 | 22703 | 7756 | nd | 16248 | 495 | 2148198 |
| Wall H93 | Wall | 5325 | 40148 | 9431 | 42759 | 4625577 | 3725 | 575 | 20850 | 226301 | 11278 | nd | 25241 | 637 | 5011847 |
| Wall H103 | Wall | 25660 | 31367 | 8240 | 43951 | 3791428 | 2928 | 556 | 7119 | 136956 | 11127 | nd | 48381 | 2602 | 4110315 |
| Wall H104 | Wall | 11117 | 6584 | 2122 | 15663 | 836926 | 2107 | 448 | 6015 | 86655 | 8276 | nd | 33471 | 2639 | 1012023 |
| Wall M1 | Wall | 3575 | 37142 | 9588 | 33161 | 4381731 | nd | 569 | 4491 | 48919 | 11228 | nd | 28527 | nd | 4558931 |
| Wall M2 | Wall | 2642 | 25297 | 6998 | 24690 | 3009549 | 255 | 561 | 6446 | 49914 | 8996 | nd | 19883 | 5642 | 3160873 |
| Wall Tect | Wall | 9176 | 68274 | 17737 | 70792 | 8173522 | 2342 | 562 | 16821 | 294786 | 15158 | nd | 75775 | nd | 8744945 |
| Wall Line | Wall | 6157 | 57896 | 15491 | 53379 | 6980723 | 2008 | 563 | 7015 | 88391 | 12641 | nd | 53953 | 1130 | 7279347 |
| Wall Sign | Wall | 4193 | 34876 | 9321 | 31192 | 4121779 | nd | 566 | 4992 | 77484 | 8643 | nd | 33976 | 4460 | 4331482 |
| Wall red | Wall red | 1627 | 8415 | 4143 | 31530 | 1385205 | 14966 | 576 | 1E+06 | 754134 | 18799 | 50330 | 12742 | 31808 | 3655717 |

intensities between the paintings and the cave wall is small in most cases. The pictorial layers are thin despite their bright colours. Additionally, the composition of the cave wall is highly variable (Fig 6B). As a consequence, no or few differences in pigment composition are detectable between representations of different themes or cultural attributions when doing direct comparison of the spectra (Fig 6A). Only one disk shows significantly higher Mn and As contents (H15). These results are consistent with those obtained on the disks of the Galería de los Discos [4].

**4.2.4 PCAs on raw and normalised spectra.** The PCA of the raw counts from all spectra provides similar results: the composition of the cave wall is highly variable and the paintings are composed of Fe-bearing pigments (PCA1; Fig 7A). However, the XRF signal of several paintings cannot be distinguished from the signal of the cave walls. This is partly due to the contribution of Fe and Ca peaks in Comp2, but also to global differences in intensity between the measurements according to Comp1 (positive contribution for the whole spectrum). In addition, results obtained on a single figure feature substantial discrepancy. Several factors can explain these patterns: mass absorption effects (Fe versus Ca-rich matrix, difference in the thickness if the pictorial layer), the presence of a layer of air, and differences in the thickness of the layer of air (see e.g. [46]). Field observations show that the paintings with a very low pXRF signal are frequently thin and/or covered by a thick calcite layer (mostly figures and hand stencils, Fig 7A). This is in agreement with a combination of mass absorption and "air" effects. The contribution of Ar K-alpha peak to the results was not as significant, as we would expect if the thickness of the air layer is the main factor involved.

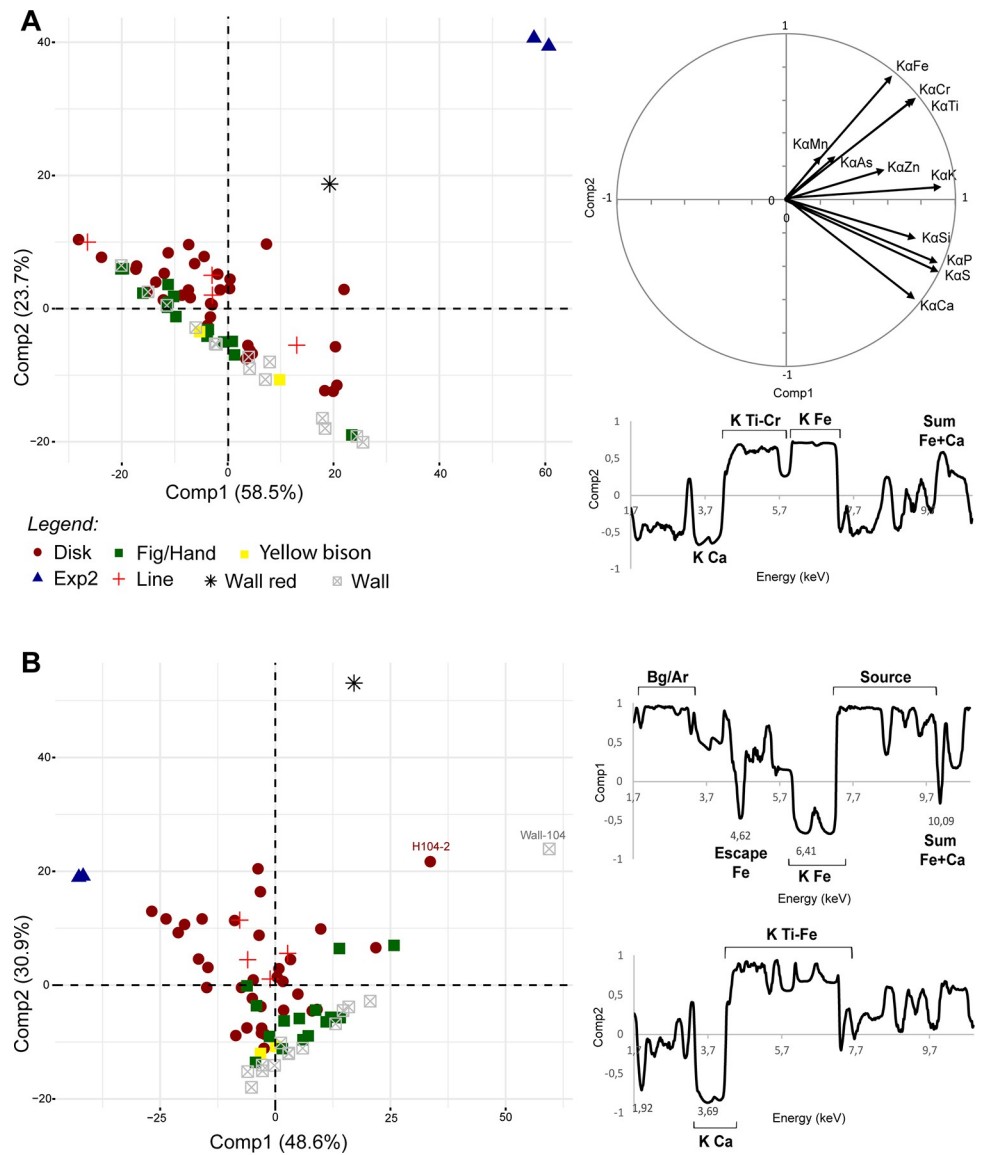

**Fig 7.** Results of PCA1 and PCA2 carried out on the raw spectra (A) and normalised spectra (B). On the right: coordinates of individuals; on the left: coordinates of variables.

Two normalisation methods were designed to counterbalance the overall differences between the spectra. The PCA of the spectral data normalised by total counts allows more effective separation between the signal of the paintings and the cave walls (PCA2; Fig 7B). The experimental painting ('Exp2'), the reddish part of the cave wall ('Wall red'), one measurement from disk 'H104' and the cave wall near this disk are separated from the cluster of other measurements. Fe peaks account for a high proportion in Comp1, in contrast with the signal of the background and the source. For Comp2, we can observe an opposition between Ca peaks on one hand, and Ti, Cr, Mn and Fe peaks on the other. However, there are irregularities in the contribution of the base and the summit of the peaks. This suggests that 'Wall-red' is rich in transition metals from Ti to Cr, 'Exp2' is enriched in Fe, while 'H104-2' (disk 'H104') and 'Wall-H104' are anomalies caused by a difference in background intensity. This is consistent

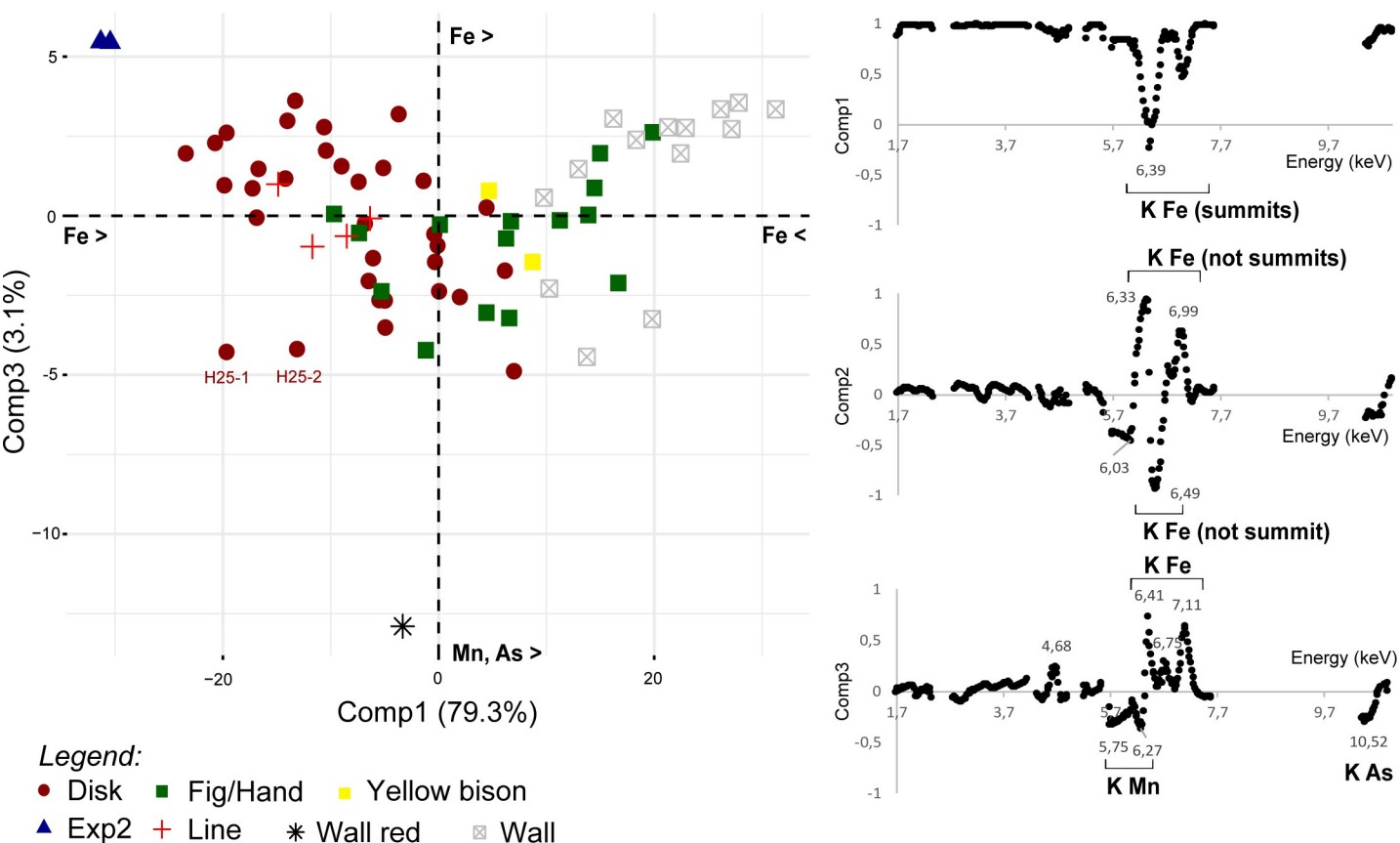

**Fig 8. Results of PCA4 of 'Fe-normalised' spectra (log of Fe ratios).** On the right: coordinates of individuals; on the left: coordinates of variables.

with a difference in the proportions of light (Ca) and heavy (Fe) elements and suggests that this data processing is impacted by matrix effects.

When Fe-ratio logarithms are used to normalise the data, the multivariate analyses allow higher discrimination between clusters of measurements (PCA4, Comp1 and 3; Fig 8). More paintings cluster separately from the cave wall, with the measurements on the 'Signal', hand stencils 'M1', 'M2' and 'M3' being the exceptions. The cave wall points are not aligned anymore, which suggests that the difference in global intensity between the spectra was significantly smoothed. The signals of 'Exp2', 'Wall red', and disk 'H25' become clearly separated from the rest of the measurements. The variability in Comp1 is explained exclusively by variations in Fe contents (summits of K peaks) and accounts for almost 80% of the total variability (Fig 8). Comp2 is explained by variations is the shape of Fe peaks. As we do not know what these differences might reflect in terms of elemental composition, we did not use this component. Comp3 only accounts for 3% of the variability, relating to variations in Fe and other elements contents: Mn (summits and other parts of K peaks) and As (summit of the K-alpha peak). The experimental paints were characterised by the highest Fe contents, with small Mn and As proportions. This is consistent with the composition of the Blutstein pigment. The results of PCA4 suggest that the 'Wall red' measurement is the richest in Mn and As, which is consistent with its spectrum (Fig 6B). The distinction of the disk H25 from other disks and figures could be due to higher Fe and As contents.

**4.2.5 PCA on peak areas (fundamental parameters, netto counts).** The PCA of netto count clr coordinates allows most of the wall cave measurements to be distinguished from the paintings, except 'Wall-H104' and 'Wall-M2' (PCAa; Fig 9A; raw data given in Table 5). For

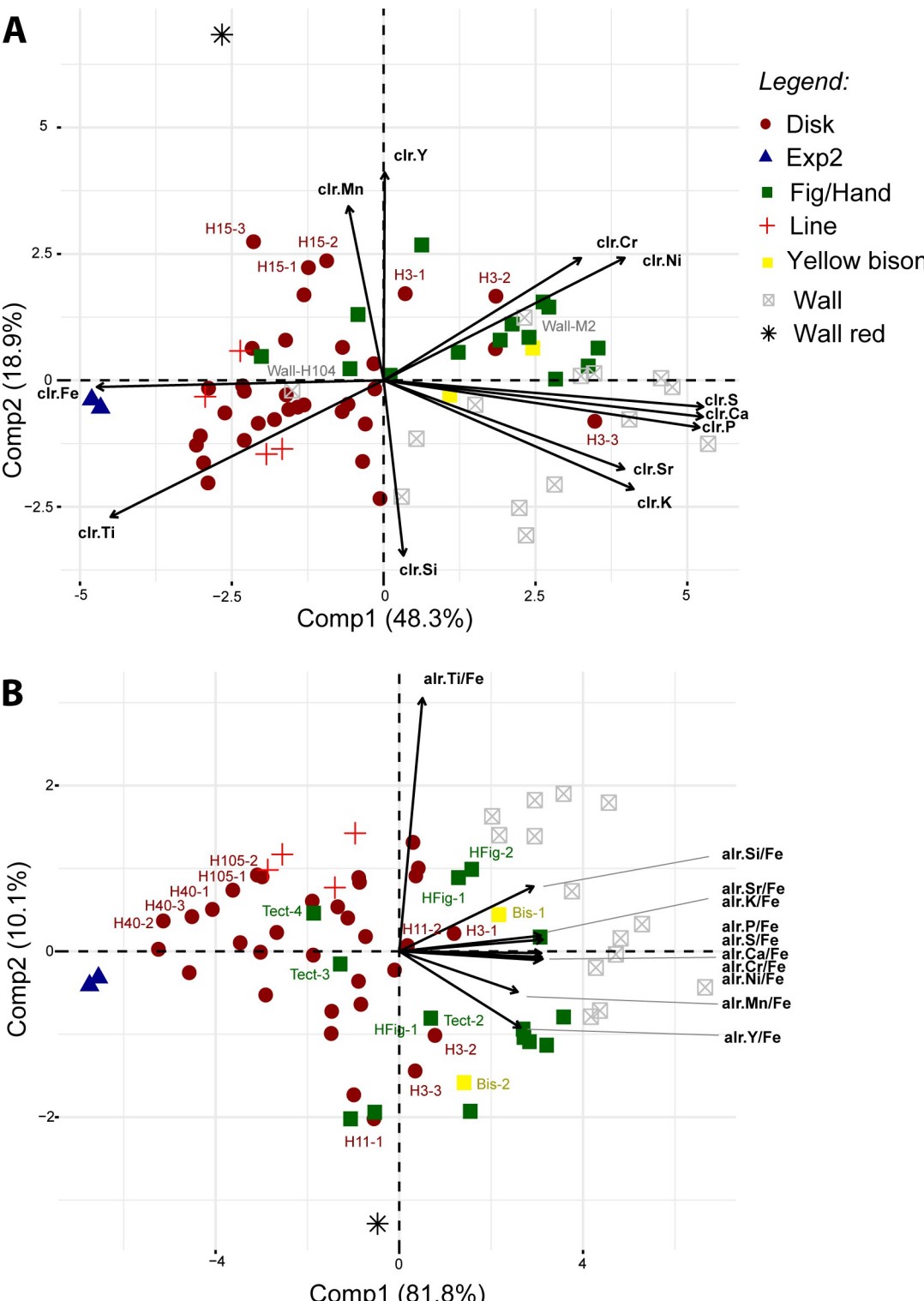

**Fig 9.** Results of PCAa and PCAc of clr coordinates (A) and alr Fe-ratios (B) of netto counts (peak areas). On the right: coordinates of individuals; on the left: coordinates of variables.

the former, the difference could be due to lower netto counts of Ca, while the latter suggests higher netto counts of Ni and Cr peaks. The measurement 'H3-3' (disk H3) is not separated from the cave wall. This could be due to higher netto counts of S and Ca peaks. This data processing is apparently subject to different biases, linked to the very small differences in Ca peak intensities between the paintings and the cave wall, but also possibly to an underestimation of matrix effects (Fe-rich matrix) and/or the impact of the X-ray source on the overall signal (Ni, Cr). One disk (H15) is partially separated from the other paintings, likely because of its higher Mn contents (Fig 6A).

PCA of the Fe-ratio alr coordinates allows complete separation of the painting and cave wall points (PCAb; Fig 9B). The 'Wall-red' measurement was separated from all the other points, as well as from the experimental painting. In most cases, all the points measured on a painting are clustered together ('Line', 'HFig', disk 'H105' or disk 'H40'), but there are several exceptions ('Tectiform', 'Bison', disk 'H11' or disk 'H3'; Fig 9B). Comp1 accounts for 82% of the variability and is characterised by a positive contribution of all elements excepting Ti/Fe, the contribution of which was almost null. Comp2 is characterised by a positive contribution of Ti/Fe (high) and Si/Fe (low) and a negative contribution of Mn/Fe and Y/Fe. We noticed that the removal of the Ni and Cr (elements that were problematic in PCAa), and P and S (elements that are highly correlated with Ca) alr coordinates did not change the main results of PCAb. The separation between the paintings and cave wall measurements seems to be due to differences in the Fe-ratios of all the elements except Ti, with a higher contribution of Si. This pattern means that the Fe proportion is significantly higher in the paintings, that of Si significantly lower, and the proportion of Ti is likely variable in both. The proportion of Mn and Y to Fe is also variable. The high dispersion of measurements from the cave wall suggests that the high variability in Ti, Mn and Y within the paintings relates to variations in the cave wall beneath them.

**4.2.6 Focus on the relation between cave wall and painting signals.** In order to test the hypothesis that most of the variability we observed within the paintings was due to high heterogeneity in cave wall composition, a final PCA was performed with Fe-ratio alr coordinates, but removing the individuals with extreme coordinates in PCAb (Exp2, Wall-Exp2 and Wall-red; PCAc; Fig 10). When the relationship between the measurement for a painting and its corresponding substrate is represented, the influence of the former on the result is quite clear: the higher Comp2 for the cave wall, the higher it is for the corresponding painting.

## 5. Discussion: Towards an understanding of the results from El Castillo *in situ* analyses

Differences in elemental composition among pigments can theoretically be interpreted as 1) variations in rock substrate composition; 2) variations in the nature and thickness of alteration deposits, including calcification layers, above and beneath the pictorial layer; 3) differences in pigment composition; 4) variations in pigment composition induced by secondary chemical processes, such as element migrations and recrystallization; or 5) repaints on the original paintings [4, 12, 36]. However, the reproducibility of pXRF analyses depends on the analytical conditions: variations in these conditions may also lead to variations in raw spectral data.

The first problematic result we obtained from the El Castillo pXRF analyses was a systematic relationship between the variations in Ca content and the overall intensity of the signal (PCA1, raw spectral data): higher Ca peak intensity corresponded to lower overall spectra intensity. This phenomenon can be explained by the thin layer of air between the instrument and the wall: its presence leads to a decrease in the signal of light elements (Ca in our case). By contrast, the signal of heavy elements (Fe in our case) is almost not impacted. Another

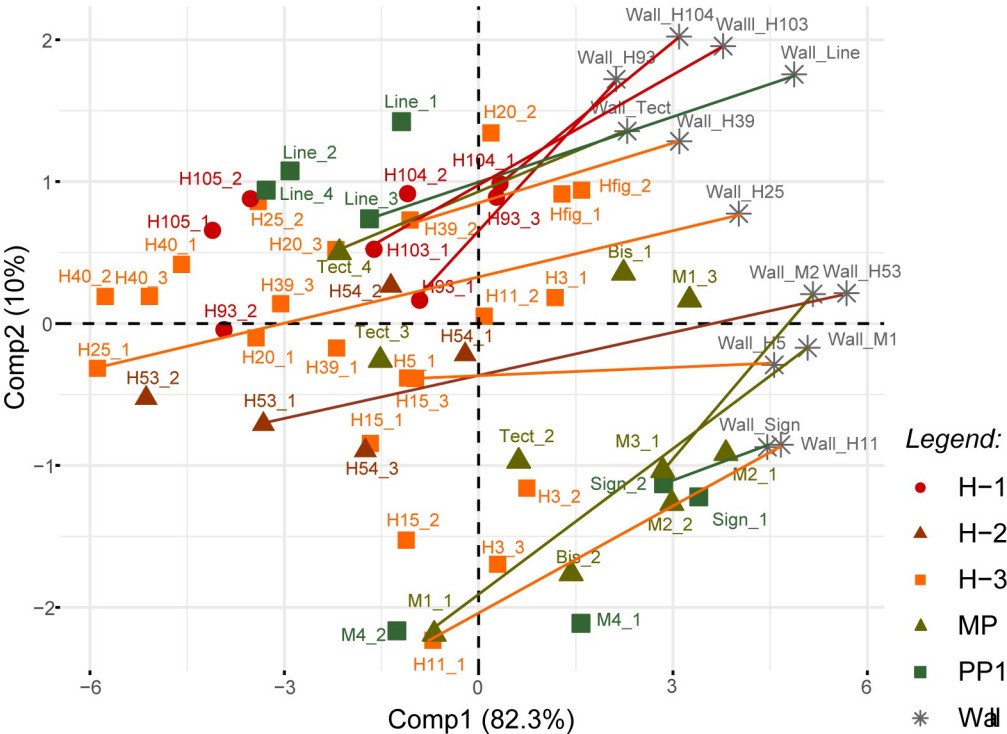

**Fig 10. Results of PCAc of alr Fe-ratios of netto counts (peak areas), measurements 'Exp2', 'wall-Exp2' and 'wall-red' excluded.** On the right: coordinates of individuals; on the left: coordinates of variables. The lines represent the vector of the 'compositional' distance between a painting and its substrate.

consequence of this phenomenon is that when spectral data are normalised by total counts, high variations in the background are observed (PCA2, normalised spectral data). In addition, this phenomenon can have an impact on the calculation of the peak areas when using the fundamental parameter method: an anomaly in Ca and Fe contents was clearly involved in the presence of an outlier within the signals of the substrate in PCAa ('Wall-H104'). Another bias was observed in the El Castillo pXRF raw signals: the influence of the Fe-matrix and/or the influence of the signal of the source (PCAa). However, when using Fe-normalisation (PCA4, b and c), all the biases virtually disappear. They may still have an impact on the results, but if there is one, it is not detected anymore. The single bias identified in PCA4, relating to the presence of variations in Fe peak shapes could be avoided by excluding the involved component (Comp2 not taken into account).

With these analytical biases addressed, the El Castillo results can now be discussed on a sounder basis. Considering 1) the high variations observed between the measurements for the substrate (experiments and all data processing), 2) the small difference between the cave wall and painting signals (all data processing except PCAb and c), 3) the absence of any differences between the yellow bison and red representations (all data processing), 4) the lack of clustering for measurements done on the same painting (all data processing) and the relationship between the signal of a painting and its corresponding cave wall measurement (PCAc), we hypothesise that most of the differences in elemental composition detected between the paintings come from variations in their environment. The composition of the rock substrate and/or the deposits on its surface, below and above the paintings, vary from place to place in the cave. The measurements on the fragments of the cave wall show that there are large differences in composition between the inside of the rock substrate and its surface. The surface is depleted in

K, Ti and As. This could suggest that the variable results we obtained mostly depend on variations in the nature and thickness of the coating deposits.

The results of PCAb support this hypothesis. The measurements of a single painting formed a cluster when Ti content was high (high Comp2 coordinate). Ti being mostly part of the inner part of the substrate, this means that when the deposit layer is thinner, variations between close measurements are smaller. The composition of this alteration layer is another important factor explaining the dispersion of the measurements. In particular, Mn rate plays a role in the variability within cave wall and painting signals according to PCAb and to the significantly higher Mn content in the reddish part of the cave wall ('Wall-red'). This is in agreement with previous observations on the composition of disk H58 (see micro-sample CAST-ADN2 in [4]): the pictorial layer shows very variable Mn content. Natural migration of Mn at the surface of the cave wall probably occurred at places, as commonly observed in karst systems [112, 113]. Y contents also appear responsible for large variations between the analyses of a single painting (PCAb). It may have a similar behaviour to Mn or be part of a specific phase coating at places the rock substrate (clay minerals?). As a consequence, Mn and Y contents cannot be used to discriminate paint pots. The behaviour of other elements is hard to establish without more detailed data on the substrate and its alteration deposits.

Regarding the identification of paint pots within El Castillo paintings, the identified clusters isolate only three paintings: the experimental painting; disk 'H15' and disk 'H25'. All the results suggest that the experimental painting was mainly distinct from the others because of higher Fe content. This is in agreement with its composition, thickness, and the absence of deposits on its surface. Disk 'H15' is richer in Mn. We have just shown that we cannot use this element to discriminate between paint pots with any reliability. This leaves us with disk 'H25' isolated by PCA4. The result of the PCA suggests that it features a higher As to Fe proportion. This element could not be used in PCAb (because of the absence or very low As peak in most spectra). But when we look at the netto counts, the high As content in 'H25' is noticeable. 'H25' is the only disk we analysed from a vertically-aligned cluster, perpendicular to the main horizontal alignment of disks. It is possible that it was not made with exactly the same paint pot as the horizontally-aligned disks, including 'H3', 'H5', 'H11', 'H20' to its left, and 'H39' and 'H40' to its right. As is potentially a trace element that can discriminate El Castillo red paintings. Although our results do not allow us to go any further in the identification of paint pots, they nevertheless provide another line of evidence supporting the hypothesis that almost-pure Fe oxide was used to make the disks in the H-sector, as micro-analyses previously suggested.

The combination of pXRF and microscopic results adds some more lines of evidence. Pigments from disks H103 and H105 have a different thickness and the presence of small black and transparent inclusions in H103 also indicates a difference in composition. They were clearly separated by PCA4 and PCAb and c (Fig 10). By increasing the number of microscopic observations, we may get a better idea of the impact of thickness and composition of pictorial layers on pXRF measurements.

## 6. Conclusion: Evaluation of *in situ* analyses and implications for future studies

To sum up the discussion of the above, our results show that secondary alteration deposits covering the cave walls are highly variable at El Castillo cave. This considerably affects the pXRF spectra of the paintings. Data processing of both the spectra and peak areas identified only one element possibly reflecting differences in paint preparation. At first sight, pXRF analyses appear to be of little interest in the study of cave painting technology. However, El Castillo limestone is rich in elements such as Si, Ti, K and even As, that are precisely those that allowed

the two red pigments used in our experiments to be identified as different paints. It remains possible that El Castillo cave constitutes a special case and that, elsewhere, the composition of the rock substrate may turn out not to represent such a limitation for the discrimination of red paint pots. By identifying the composition of the alteration deposits on the cave wall without any sampling, we were able to establish which paintings are the most affected by these deposits. This information will be precious if further micro-sampling of the paintings is made possible.

The El Castillo case illustrates once again that all forms of analyses are precious for the understanding of paint preparation techniques in rock art: 1) analysis of the paintings themselves, but also analysis of the substrate and alteration deposits; 2) *in-situ* analyses as presented here, but also laboratory analyses of micro-samples. In sum, we do advocate use of pXRF analyses, bearing in mind that they are not a replacement for the conventional laboratory analyses that remain the main way to validate the hypotheses put forward on the basis of *in situ* surface analyses. We are in favour of a more systematic evaluation of the impact of cave wall heterogeneity on *in situ* pXRF analyses. For instance, at Font-de-Gaume, the composition of the substrate and alteration deposits is not discussed [36]. There is no guarantee that the 'blank measurements' that were taken offset the heterogeneity of the cave wall. Differences in Mn content in the red paints of the bison could be due to heterogeneous Mn contents in alteration deposits, as observed at El Castillo. Similarly, the absence of comparison between the paintings at Le Peña and their environment entails that we cannot be sure of the origin of the differences observed between them [34]. We have shown that pXRF measurements have the potential to generate more accurate hypotheses if cave wall measurements are evaluated with more care.

It is worth highlighting, however, that we also identified significant analytical biases in the pXRF results that do not depend on the site nor on the instrument used. Considering the large variations in Ca content between the measurements, the thin layer of air between the incident X-ray beam and the surface being analysed have an important effect on the spectra. This effect may not be entirely corrected by the fundamental parameters method. When contact between the instrument and the paintings is not possible, the use of a flow of Helium would avoid such a bias. Unfortunately, such equipment is difficult to transport inside a cave. For this reason, the step-by-step method of enhanced data processing that we applied in this study presents considerable advantages. Multivariate analyses of the spectra are key for the identification of all types of biases and Fe-normalisation smooths them out considerably. Some authors prefer to rely on semi-quantitative data [5, 20, 36, 83], but this 'blind' method was not suitable in our case, considering the large variations in Fe and Ca contents. Matrix effects are too strong in such conditions [46, 114, this study]. Future research in rock paintings would greatly benefit from the creation of dedicated references allowing the calibration of pXRF instruments for the quantification of elements in Fe-bearing paints on limestone substrates, following on from the calibration tests carried out recently for Fe-based pigments [101].

The final conclusion we can draw from our study is that the interest of microscopic analyses is underestimated. They do take time, which is why we were not able to do them systematically, but when we consider the difficulty of identifying, whatever the context, paint preparation techniques by pXRF analyses, and the difficulties in getting micro-sampling authorisations, the interest of more systematic *in-situ* microscopic examination of rock paintings becomes evident.

## Supporting information

**S1 File. Location of pXRF analyses.**
(PDF)

**S1 Table. Counts of pXRF measurements (raw spectra).**
(XLSX)

## Acknowledgments

We would like to thank Garrido Pimentel and Raul Gutiérrez (Cuevas Prehistóricas de Cantabria) for guiding us at El Castillo cave, as well as the *Consejería de Educación*, *Cultura y Deporte* of the Government of Cantabria for granting us permission to work and conduct *in situ* analyses at the site. Thank you to Catherine Ferrier (Université de Bordeaux, UMR PACEA) for showing us the use of the digital microscope and giving us a block of Rupelian limestone from her own research. We also thank Eric Pubert (UMR PACEA) for building the support for the pXRF instrument and to Alain Queffelec (UMR PACEA) for discussion on the choice of pXRF analytical parameters. Our thanks also go to Hélène Salomon (UMR EDYTEM) for discussions on pXRF *in situ* analyses.

## Author Contributions

**Conceptualization:** Laure Dayet, João Zilhão.

**Data curation:** Laure Dayet.

**Formal analysis:** Laure Dayet.

**Funding acquisition:** Francesco d'Errico.

**Investigation:** Laure Dayet, Francesco d'Errico, Marcos García Diez, João Zilhão.

**Methodology:** Laure Dayet.

**Project administration:** Francesco d'Errico, João Zilhão.

**Resources:** Francesco d'Errico.

**Supervision:** Francesco d'Errico.

**Writing – original draft:** Laure Dayet.

**Writing – review & editing:** Francesco d'Errico, Marcos García Diez, João Zilhão.

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
