## [Decision Letter · Decision Letter 0]

18 Oct 2021

PONE-D-21-22143Critical evaluation of in situ analyses for the characterisation of red pigments in rock paintings: a case study from El Castillo, SpainPLOS ONE

Dear Dr. Dayet,

Thank you for submitting your manuscript to PLOS ONE. After careful consideration, we feel that it has merit but does not fully meet PLOS ONE’s publication criteria as it currently stands. Therefore, we invite you to submit a revised version of the manuscript that addresses the points raised during the review process.

Reviewers appreciated your work and suggested for minor revisions. Interviewing your manuscript, please take care of all comments. Also, I would be happy if you can explain the major implications on the use of p-XRF, as suggested by one of the reviewers.==============================

We look forward to receiving your revised manuscript.

Kind regards,

Andrea Zerboni, Ph.D.

Academic Editor

PLOS ONE

2. In your manuscript, please provide additional information regarding the specimens used in your study. Ensure that you have reported specimen numbers and complete repository information, including museum name and geographic location.

For more information on PLOS ONE's requirements for paleontology and archaeology research, see https://journals.plos.org/plosone/s/submission-guidelines#loc-paleontology-and-archaeology-research.

“We would like to thank Garrido Pimentel and Raul Gutiérrez (Cuevas Prehistóricas de Cantabria) for guiding us at El Castillo cave, as well as the Consejería de Educación, Cultura y Deporte of the Government of Cantabria for granting us permission to work and conduct in situ analyses at the site. Thank you to Catherine Ferrier (Université de Bordeaux, UMR PACEA) for showing us the use of the digital microscope and giving us a block of Rupelian limestone from her own research. We also thank Eric Pubert (UMR PACEA) for building the  support for the pXRF instrument and to Alain Queffelec (UMR PACEA) for discussion on the choice of pXRF analytical parameters. Our thanks also go to Hélène Salomon (UMR EDYTEM) for discussions on pXRF in situ analyses. This research has been supported by a grant from the European Research Council (FP7/2007/2013, TRACSYMBOLS 249587). The work of Francesco d’Errico is also supported by the Programme Talents and the Grand Programme de Recherche Human Past of the University of Bordeaux Initiative of Excellence, and the Research Council of Norway through its Centres of Excellence funding scheme, SFF Centre for Early Sapiens Behaviour (SapienCE), project number 262618. PACEA (UMR5199 CNRS).

We note that you have provided funding information that is not currently declared in your Funding Statement. However, funding information should not appear in the Acknowledgments section or other areas of your manuscript. We will only publish funding information present in the Funding Statement section of the online submission form. “

“The work of Laure Dayet was supported by a grant from the European Research Council (FP7/2007/2013, TRACSYMBOLS 249587). The work of Francesco d’Errico is also supported by the Programme Talents and the Grand Programme de Recherche Human Past of the University of Bordeaux Initiative of Excellence, and the Research Council of Norway through its Centres of Excellence funding scheme, SFF Centre for Early Sapiens Behaviour (SapienCE), project number 262618. PACEA (UMR5199 CNRS).”

Additional Editor Comments (if provided):

Reviewers' comments:

Reviewer's Responses to Questions

**Comments to the Author**

1. Is the manuscript technically sound, and do the data support the conclusions?

Reviewer #1: Yes

Reviewer #2: Yes

2. Has the statistical analysis been performed appropriately and rigorously? 

Reviewer #1: I Don't Know

Reviewer #2: Yes

3. Have the authors made all data underlying the findings in their manuscript fully available?

Reviewer #1: Yes

Reviewer #2: Yes

4. Is the manuscript presented in an intelligible fashion and written in standard English?

Reviewer #1: Yes

Reviewer #2: Yes

5. Review Comments to the Author

Reviewer #1: The work described in this manuscript demonstrates the limitations of handheld XRF measurements applied to the analysis of cave rock art.

The conclusions -- essentially that pXRF measurements have some value in surveying the elemental composition of paint, substrate, and alteration deposits but, in the context of rock art, cannot be relied upon to provide definitive, quantitative answers -- will not come as a surprise to XRF experts. The limitations of pXRF have been established previously for other archaeological materials, notably glasses and ceramics. In these studies, the use of reference standards, improved data collection protocols, and improved processing has improved data quality considerably.

As the authors point out, the situation for understanding rock art is made more challenging by common prohibitions against sampling or destructive analyses, leaving only a few nondestructive, field-based techniques like pXRF available to researchers.

A significant issue continues to be non-experts who do not appreciate the limitations of pXRF and thus ultimately over-interpret results (as scientists, authors, and reviewers). This situation certainly is not unique to XRF but can be seen for other analytical techniques including EDS, XPS, etc. Although advances in technology have made the mechanics of pXRF data collection easy, the critical interpretation of data leaves something to be desired in many cases.

The work described in this manuscript is a valuable contribution to the rock art field because it clearly shows the sources of common pXRF errors and the special circumstances and challenges related to heterogeneous, ancient paintings in cave environments.

Recommend publication after minor changes:

-- pXRF should be defined earlier, at its first appearance

-- some information and parallels from the literature describing XRF analysis of glasses and ceramics should be incorporated in this manuscript

Reviewer #2: Dear colleagues,

It was a pleasure to read this article by Dayet et al. on their p-XRF analysis at the El Castillo cave. This article is well-written, easy to follow and present a very important contribution to the use of p-XRF for rock art analysis. The processing of the data with different methods with the aim to understand biases that may be introduced by this method is very welcome at a time when such a method is now widely used without the required cares. Although focusing on the rock art case, the results presented here are relevant for the study of archaeological materials with p-XRF in general.

All data acquired through this study are available in supplementary materials, which allow any researcher to reproduce the processing. I thank the authors for this.

In conclusion, I recommend this article to be accepted with minor revisions listed below.

Please see the attached file for detailed comments.

6. PLOS authors have the option to publish the peer review history of their article (what does this mean?). If published, this will include your full peer review and any attached files.

Reviewer #1: No

Reviewer #2: No

---

## [Author Response · Author response to Decision Letter 0]

6 Dec 2021

Reviewer #1: The work described in this manuscript demonstrates the limitations of handheld XRF measurements applied to the analysis of cave rock art.

The conclusions -- essentially that pXRF measurements have some value in surveying the elemental composition of paint, substrate, and alteration deposits but, in the context of rock art, cannot be relied upon to provide definitive, quantitative answers -- will not come as a surprise to XRF experts. The limitations of pXRF have been established previously for other archaeological materials, notably glasses and ceramics. In these studies, the use of reference standards, improved data collection protocols, and improved processing has improved data quality considerably.

As the authors point out, the situation for understanding rock art is made more challenging by common prohibitions against sampling or destructive analyses, leaving only a few nondestructive, field-based techniques like pXRF available to researchers.

A significant issue continues to be non-experts who do not appreciate the limitations of pXRF and thus ultimately over-interpret results (as scientists, authors, and reviewers). This situation certainly is not unique to XRF but can be seen for other analytical techniques including EDS, XPS, etc. Although advances in technology have made the mechanics of pXRF data collection easy, the critical interpretation of data leaves something to be desired in many cases.

The work described in this manuscript is a valuable contribution to the rock art field because it clearly shows the sources of common pXRF errors and the special circumstances and challenges related to heterogeneous, ancient paintings in cave environments.

Recommend publication after minor changes:

-- pXRF should be defined earlier, at its first appearance

-- some information and parallels from the literature describing XRF analysis of glasses and ceramics should be incorporated in this manuscript

Reply

We thank this reviewer for his/her appreciation of our work and for underling its methodological interest.

We have introduced the two changes suggested by this reviewer.

As regard to the first suggestion of change, the manuscript was changed as follows:

Line 34: “In-situ analyses performed with portable X-ray fluorescence (pXRF) and Raman spectroscopy equipments are becoming widespread for the study of paint materials and painting techniques.”

As regard to the second suggestion, our article is dedicated to rock art issues. Although we develop pXRF data treatments, this was not the main aims of our paper. We nonetheless added one sentence:

Line 84: “The interest and limitations pXRF analysis has already been discussed for other archaeological applications (see e.g. Shackley 2011; Speakman et al. 2011; Vazquez et al. 2012; Speakman and Shackley 2013; Wilke 2017; Tykot 2021) but no in-depth studies have been carried out in the critical domain of painted rock art.”

 

Reviewer #2: Dear colleagues,

It was a pleasure to read this article by Dayet et al. on their p-XRF analysis at the El Castillo cave. This article is well-written, easy to follow and present a very important contribution to the use of p-XRF for rock art analysis. The processing of the data with different methods with the aim to understand biases that may be introduced by this method is very welcome at a time when such a method is now widely used without the required cares. Although focusing on the rock art case, the results presented here are relevant for the study of archaeological materials with p-XRF in general.

All data acquired through this study are available in supplementary materials, which allow any researcher to reproduce the processing. I thank the authors for this.

In conclusion, I recommend this article to be accepted with minor revisions listed below.

Review for « Critical evaluation of in situ analyses for the characterisation of red pigments in rock paintings: a case study from El Castillo, Spain », Dayet et al. 

Dear colleagues, 

It was a pleasure to read this article by Dayet et al. on their p-XRF analysis at the El Castillo cave. This article is well-written, easy to follow and present a very important contribution to the use of p-XRF for rock art analysis. The processing of the data with different methods with the aim to understand biases that may be introduced by this method is very welcome at a time when such a method is now widely used without the required cares. Although focusing on the rock art case, the results presented here are relevant for the study of archaeological materials with p-XRF in general. 

All data acquired through this study are available in supplementary materials, which allow any researcher to reproduce the processing. I thank the authors for this. 

In conclusion, I recommend this article to be accepted with minor revisions listed below.

Major comments

-Lines 97 to 116: This paragraph is very interesting and reflects the choices that are constrained by the use of raw materials and technics of applications. However, I would emphasis that a consideration is missing here: the type of binders chosen. Indeed, your binder has to be chosen in adequation to your raw materials, its texture and granulometry, and to the application technique. Although you mention the paint materials at large, I feel it may be needed to emphasise that both the colouring materials and the binders are of importance. I would also suggest elaborating a little more on “technical facts” (predictable gestures) and “degrees of facts” (unpredictable gestures linked to social behaviours): distinctions from Leroi-Gourhan’s theory on the “chaîne opératoire”. Indeed, the materials chosen, their preparation and the application techniques may be related to specific spiritualities or rituals and gestures adapted to fit the use of such materials/techniques. 

Reply:

We thank the reviewer for underling the potential role of binders and cultural choices as factors influencing the paint production and application. We have changed our text as following:

Line 109: “In this regard, the choice of the binder is also an important parameter. Technical choices may be limited by raw material availability, depending on a geological context for the inorganic part of the paint mixture.”

Few sentences later we have added the following sentence:

Line 114: “Finally, binders and pigments are not independent parameters. The choice of one influence the choice of the other. They are also highly influenced by a society’s overall technical system (e.g. use of the same binder for different purposes) and the way in which the painter has been taught to mix and apply the paint. In other words, the cultural logic driving the “chaîne opératoire” (Leroi-Gourhan 1964) may play a key role in the final appearence of the painting.” 

-Lines 118 to 137: The changes in the raw materials/paint recipes may also be related to trade networks and to exchanges of believes/behaviours. We may also consider that several groups with different behaviours came to paint in the same cave at the same “period”. The dating methods do not allow a very precise chronology for such very old periods, with nomadic groups moving in large areas. Hence, different paint recipes may only be related to several groups painting in the same cave with different techniques. 

ReplyWe added a couple of sentences at the end of the paragraph to incorporate the point made by this reviewer.

Line 149: “In addition, paint recipes may vary as a function of changes in trade networks and patterns of cultural exchange. It is also possible that groups with different artistic practices came to paint in the same cave roughly during the same period. Available dating methods do not allow for the chronological precision required to investigate such sources of variation.”

-Although you were not able to analyse the same figures that d’Errico et al. 2016, I feel like links between the two are missing. A table may be added that summarize similarities and differences between your study and d’Errico et al. 2016. 

Reply

This link is given in table 2. In this table we establish a connection between the disks from which come the micro-samples analysed in our 2016 papers and the disks studied in the present work.

-You included a yellow paint as part of your study but you stated several times that you will focus on the red paintings (line 79 for example). But at the end, you mention that you used the yellow painted figure to determine if it is distinguishable from the red paintings. I feel that this information should come at the beginning of the article and you may also state why you chose only one yellow paint: accessibility, thickness of the layer, chronology, other reason? This yellow painting is also not visible in your PCA figures. Was it considered? If so, I would suggest marking it with a different bullet in your figures. It will help the reader. 

Reply

Only one yellow painting was analysed because yellow paintings are rare in the cave and most of them are very thin. In the revised version of the manuscript, we mention to the yellow painting in the introduction, as requested by this reviewer.

Line 81: “. To achieve this, we carried out microscopic examination and pXRF analyses of several red paintings and one yellow painting for comparison purposes, and used experimental paintings and statistical analyses to assess the reliability of the results obtained on the El Castillo paintings.”

All the figures were modified accordingly: the yellow painting is now represented with a different symbol on the binary diagrams.

-At lines 645-646, you quickly mention the variations within one figure but it is not visible in the rest of your paper and in your figures. I feel it would be useful to have more information regarding the reproducibility of the measures on one single figure to better appreciate the issues of differentiation between several figures. I would suggest adding a quick paragraph on this and maybe emphasising them for a couple of relevant examples in the figures. 

Reply

In order to show that we did take into account this parameter, one sentence was added in ther result section at the beginning of sub-section dealing with PCAs.

Line 520: “In addition, results obtained on a single figure may feature substantial discrepancies. Several factors can explain these trends:”

I would like to emphasis a few more references for your study (for p-XRF analysis, and for the recognition of paint pots and their chronological implications):

- MacDonald, B. L., et al. (2019). "Hunter-Gatherers Harvested and Heated Microbial Biogenic Iron Oxides to Produce Rock Art Pigment." Scientific Reports 9(1): 1-13.

- MacDonald, B. L. (2015). Methodological developments for the geochemical analysis of ochre from archaeological contexts: Case studies from British Columbia and Ontario, Canada. Anthropology, McMaster University. PhD: 209.

- Trosseau, A., et al. (2021). "In-situ XRF study of black colouring matter of the Palaeolithic figures in the Font-de-Gaume cave." Journal of Analytical Atomic Spectrometry.

- Castañeda, A. M., et al. (2019). "Portable X-ray fluorescence of Lower Pecos painted pebbles: New insights regarding pigment choice and chronology." Journal of Archaeological Science: Reports 25: 56-71.

- Bonneau, A., et al. (2021). "Characterization and dating of San rock art in the Metolong catchment, Lesotho: A preliminary investigation of technological and stylistic changes." Quaternary International.

- Bonneau, A., et al. (2017). "A pigment characterization approach to selection of dating methods and interpretation of rock art: the case of the Mikinak site, Lake Wapizagonke, Quebec, Canada." Archaeometry 59(5): 834-851

- Sepúlveda, M. (2021). "Making visible the invisible. A microarchaeology approach and an Archaeology of Color perspective for rock art paintings from the southern cone of South America." Quaternary International 572: 5-23.

Reply

All the references were added.

Minor comments

Text:

- There are several lists of references for which I would suggest adding e.g. at the beginning. Although your review of the literature is very good, I was able to spot some more references (see above). To avoid very long reference lists, here are the lines with the reference list to which I would add e.g. at the beginning: 

o Line 31

o Line 92

o Line 147

DONE.

- Lines 66 – 67: I would suggest presenting the U-series dates as follow, which is a more conventional way to report the age:

o 41.4±0.6 ka

o 35.7±0.6 ka

DONE.

- Line 163 – 175: on the use of in-situ Raman spectroscopy, I would suggest emphasising the possibility of burnt marks if the system is not used correctly. Although, the instruments are getting safer and the spot sizes are getting smaller, Raman spectroscopy remains a “risk of damage” if not used correctly.

We agree with the point made by this reviewer but think that adding this is out of the scope of our paper, which specifically focuses on advantages and disadvantages of pXRF. If we were to follow the logic he/she proposes, we would also have to discuss the possible damages produced by other techniques such as portable FTIR or close microscopic analysis of the wall. We think that this would be out of place in an article like ours.

- Line 180 – 181: amorphous carbon cannot be “identified” with p-XRF, it can be assumed if no Mn or Fe is present in the spectrum. It is indeed hair-splitting but this sentence may give false information to non-initiate readers.

The text was modified as follow: 

Line 196: “Theoretically, several families of pigments can be identified with this equipment: iron-based yellow and red pigments, black manganese oxides, white phosphates, carbonates, titanium oxides. For black paintings, the absence of manganese and iron on the spectrum is an indication in favour of the use of pigments rich in carbon.”

- Line 330: you mentioned a red natural deposit in Sala de los Polícromos that you analysed. Is there any photo related to this natural deposit? How is it called in your data set?

We have changed out text to explain how the natural deposit is called in our dataset. The text was modified as follow: 

Line 346: “That is why the number of substrate measurements is lower than the number of paintings analysed. In the Sala de los Polícromos, a red natural deposit (Wall red), was also measured 

- Line 363: “paint pots” rather than “pot paints”.

DONE.

- Lines 707 – 711: I would suggest removing the references in your points. This conclusion presents the results of your study, which indeed confirm previous ones. However, as it is written, it looks like these conclusions are drawn from previous studies and not from this paper. You may emphasis more your own contribution to these previous studies. This paper is a really important one with the several ways of processing data and their critical evaluation.

DONE.

Tables and figures:

- Table 2: I would suggest aligning numbers of “p-XRF analyses” and “microscope” columns in the centre. It will make it more understandable. 

DONE.

- Table 3 caption: I would suggest repeating the article in which the compositional data are extracted for Ref-Fe1.

DONE.

- Fig 10: The lines are difficult to understand and “hide” a part of the graphic. I would suggest adding a table on the right of the graphic to list the correlations between the paintings and the wall analyses.

It is true that the lines partially hide a part of the graphs but we found no other way to make the reader sees the links between points.

---

## [Editor Report · Decision Letter 1]

19 Dec 2021

Critical evaluation of in situ analyses for the characterisation of red pigments in rock paintings: a case study from El Castillo, Spain

PONE-D-21-22143R1

Dear Dr. Dayet,

We’re pleased to inform you that your manuscript has been judged scientifically suitable for publication and will be formally accepted for publication once it meets all outstanding technical requirements.

Kind regards,

Andrea Zerboni, Ph.D.

Academic Editor

PLOS ONE
---

## [Editor Report · Acceptance letter]

31 Dec 2021

PONE-D-21-22143R1 

Critical evaluation of *in situ* analyses for the characterisation of red pigments in rock paintings: a case study from El Castillo, Spain 

Dear Dr. Dayet:

I'm pleased to inform you that your manuscript has been deemed suitable for publication in PLOS ONE. Congratulations! Your manuscript is now with our production department. 

Kind regards, 

on behalf of

Prof. Andrea Zerboni 

Academic Editor

PLOS ONE